



# Size-segregated characteristics of OC, EC and organic matters in PM emitted from different types of ships in China

**Fan Zhang[1,2], Hai Guo[1]\*, Yingjun Chen[2,3,4]\*, Volker Matthias[5], Yan Zhang[2], Xin Yang[2,4], Jianmin Chen[2]**

[1]Air Quality Studies, Department of Civil and Environmental Engineering, The Hong Kong Polytechnic University, Hong Kong, P.R. China

[2]Shanghai Key Laboratory of Atmospheric Particle Pollution and Prevention (LAP3), Department of Environmental Science and Engineering, Fudan University, Shanghai 200092, P.R. China

[3]Key Laboratory of Pollution Control and Resource Reuse, College of Environmental Science and Engineering, Tongji University, Shanghai 200092, P.R. China

[4]Shanghai Institute of Pollution Control and Ecological Security, Shanghai 200092, P.R. China

[5]Helmholtz-Zentrum Geesthacht, Institute of Coastal Research, Max-Planck-Straße 1, 21502 Geesthacht, Germany

**Correspondence:** Hai Guo (hai.guo@polyu.edu.hk)

Yingjun Chen (yjchenfd@fudan.edu.cn)



**Abstract**

Studies of detailed chemical compositions in particles with different size ranges emitted from ships are in serious shortage. In this study, size-segregated distributions and characteristics of particle mass, organic carbon (OC), elemental carbon (EC), 16 EPA PAHs and 25 n-alkanes measured on board of 12 different vessels in China were given. The results showed that: (1) More than half of the total particle mass, OC, EC, PAHs and n-alkanes were concentrated in fine particles with aerodynamic diameter $(D_p)$<1.1 μm for most of the tested ships, basically presenting downward distribution trends with the increase of particle size. However, different types of ships showed quite different particle size-dependent chemical compositions. (2) In fine particles, the OC and EC were the dominant components, while in coarse particles, OC and EC only accounted for very small proportions. With the increase of particle size, the OC to EC ratios first decreased and then increased, having the lowest values for particle sizes between 0.43μm and 1.1 μm. (3) OC1, OC2 and OC3 were the dominant OC fragments for all the tested ships, while EC1 and EC2 were the main EC fragment for ships running on heavy fuel oil (HFO) and marine diesel fuel, respectively; Different OC and EC fragments presented different distributions in different particle sizes. (4) Low power diesel fishing boats (LPDF) had much higher PAHs emission ratios than high power diesel vessels (HPDV) and heavy fuel oil vessel (HFOV) in fine particles, and HFOV had the lowest values. (5) PAHs and n-alkanes showed different profile patterns for different types of ships and also in different particle size bins, which meant that the particle size should be considered when source apportionment was conducted. It is also noteworthy from the results in this study that the smaller the particle size, the more toxic the particle was, especially for the fishing boats in China.

## 1. Introduction

Particulate matter (PM) emitted from ships have significant impacts on human health and air quality (Schröder et al., 2017; Liu et al., 2016; Oeder et al., 2015; Viana et al., 2014). Ship emission is one of the most important sources of fine particulate matter ($PM_{2.5}$) in harbor or offshore areas. PM from both heavy fuel oil and marine





diesel fuel shipping emissions shows strong biological effects on human lung cells (Oeder et al., 2015). A previous study indicated that shipping emissions in East Asia lead to 8,700-25,500 premature deaths per year due to $PM_{2.5}$ (Liu et al., 2016). PM emitted from ships in harbor cities or areas accounts for non-ignorable proportions of

primary $PM_{2.5}$ (Gregoris et al., 2016; Zhao et al., 2013; Agrawal et al., 2009). This fraction can reach up to 17-30% if the contribution of shipping emissions to secondary particles is considered, too (Pandolfi et al., 2011; Liu et al., 2017). The chemical composition of PM emitted from ships is complex. Organic matter (OM), elemental carbon (EC) or black carbon (BC), water-soluble ions and heavy metals are the main

components. In addition, PAHs in PM emitted from ships have drawn much attention due to their significantly negative impact on human health, as well as their diagnostic characteristics for source apportionment (Vieira de Souza and Corrêa, 2015; Pongpiachan et al., 2015; Alves et al., 2015). Ships are considered to be one of the most important sources for BC emissions in Arctic areas (Schröder et al., 2017; Quinn

et al., 2011; Corbett et al., 2010) that could lead to the increase of ice melt due to its strong light absorbing properties. BC-containing aerosols in the Arctic can perturb the radiation balance with either a warming effect or a cooling effect on climate depending on the albedo of the underlying surface relative to the albedo of the BC haze itself (Quinn et al., 2011). Besides, as a key component of soot, BC is also

considered to have substantial negative consequences for health. Furthermore, as important components, water-soluble ions such as $SO_4^{2-}$, $NO_3^-$, $NH_4^+$, $Cl^-$ and $Na^+$ are routinely studied due to their unique emission characteristics in PM of ships, different from other sources (Sippula et al., 2014; Moldanová et al., 2013; Moldanová et al., 2009; Agrawal et al., 2008). Notably, heavy metals generally have high levels in PM

emitted from ships compared to other sources. This especially holds for HFO fueled ships where metals like vanadium can be used as tracers (Moldanová et al., 2009; Agrawal et al., 2009). In addition, because a large number of particles emitted from ships are very small (<0.1 μm), which may have significant impacts on cloud formation (Fridell et al., 2008), particle number concentrations particularly for



ultrafine particles have gained more and more attention in recent years.

The size-resolved number and mass distributions of particles emitted from ship engines have been investigated for more than one decade due to their potential climate impacts and influence on human health. Three measurement methods including engine test, onboard test and plume tracking are usually reported in the literature.

However, most of the previous studies focused on the size-resolved particle number distributions (Wu et al., 2018; Cappa et al., 2014; Beecken et al., 2014; Moldanová et al., 2013; Juwono et al., 2013; Diesch et al., 2013; Alfoeldy et al., 2013; Winnes and Fridell, 2010; Kasper et al., 2007; Cooper, 2003), which reported that the total particle number concentrations were dominated by ultrafine particles (nucleation mode) or

fine particles. For example, the mean particle diameter was between 25 and 40 nm for a two-stroke marine diesel engine (Kasper et al., 2007) and between 40 and 60 nm for a four-stroke marine diesel engine measured on a test bench (Petzold et al., 2008). Unimodal characteristics with the peak at 0.1 μm (Sinha et al., 2003; Fridell et al., 2008) or 40-50 nm (Chu-Van et al., 2017) or 30-40 nm (Winnes and Fridell, 2010) or

even 10 nm (Moldanová et al., 2013) were observed during onboard tests. Bimodal patterns were found in plumes with maxima at 10-20 nm and 100 nm (Petzold et al., 2008), and 10 nm and 35 nm (Diesch et al., 2013), respectively. Particles with a marked nucleation mode (10-100 nm) (Hobbs et al., 2000; Healy et al., 2009) and 58-131 nm (Juwono et al., 2013) were also found in the plumes from different marine

fuel oil commercial ships and berthing bulk and container ships. It should be noted that the engine type and operating mode could affect the particle number distribution (Juwono et al., 2013; Cappa et al., 2014).

Few studies involved in size-resolved particle mass distributions emitted from ships (Chu-Van et al., 2017; Moldanová et al., 2013; Murphy et al., 2009; Moldanová

et al., 2009; Fridell et al., 2008). Although most of the particles from ships were concentrated in nucleation mode in terms of number concentration, the particle mass was dominated by particles in the accumulation mode or even coarse mode. For example, it was found that the mass distribution of hot-exhaust particles from a large



ship diesel engine had two main peaks: one in the accumulation mode at Dp around

0.5 μm and the other in the coarse mode at Dp around 7 μm (Moldanová et al., 2009).

Another earlier study also found that particles with a diameter of approximately 8 μm

presented the largest mass peak in the mass spectrum for three tested ships (Fridell et

al., 2008). The size-resolved mass distribution of particles has not been fully studied.

One reason is that although the coarse particles account for non-ignorable proportions

of the total mass, they are relatively few in number concentration, which has not

caught much attention. Moreover, particles with size around 8 μm are usually not

detected by particle counters when the counting method is used to calculate the

particle mass. Only one study collected particles from ships using in-stack cascade

impactor, which revealed different mass distribution patterns from the particle

counting method (Cooper, 2003). The result indicated that though the smaller particles

(<1 μm) were dominant in number concentration, they only accounted for 10-50% of

the total mass (Cooper, 2003). The counting method was simple but had drawback;

namely, the mass could bias towards larger particles.

A handful of studies on detailed chemical compositions of particles in different

sizes were conducted (Wu et al., 2018; Murphy et al., 2009). An aerosol mass

spectrometer (AMS) was used in a previous study to give organic carbon and sulfate

contents in ultrafine particles, but no black carbon and detailed molecular chemical

components could be monitored (Lu et al., 2006). In addition, in-stack ship-based

particle measurement was performed using micro-orifice uniform deposit impactor

(MOUDI) with off-line analysis. The results showed that freshly emitted particles

from ship exhaust comprised approximately 30% organic carbon and 70% sulfuric

acid by mass (Murphy et al., 2009). However, chemical compositions and their

characteristics in particles at different sizes emitted from ships are still unclear,

especially for fine or ultrafine particles. Only one study investigated detailed PAH

speciation and total-$BaP_{eq}$ in size-segregated PM in the exhaust of a container ship

with HFO and diesel oil (DO) as fuel, and found that health risks increased with the

decrease of particle size (Wu et al., 2018). Hence, it is urgent to gain detailed





chemical compositions such as OC, EC and organic matters in size-segregated particles from different ships.

In this study, size-segregated particle samples from 12 different vessels were collected using an 8-stage cascade impactor. The mass, OC, EC, 16 EPA PAHs and 25 n-alkanes in particles of each size bin were detected. Size-segregated distributions and characteristics of mass, OC, EC and the detected organic matters were given. Potential health impact, source apportionment and particle formation mechanism of ship

exhaust were also explored in this study.

## 2. Materials and methods

### 2.1 Tested vessels and fuels

Size-segregated particle samples on quartz filters from 12 different vessels were collected in this study, including one heavy fuel oil ocean-going vessel, eight fishing

boats, one engineering ship, and two research ships. Their technical parameters are shown in Table S1. These vessels were classified into three types based on their engine type and fuel type, namely, low-power-diesel fishing boat (LPDF), high-power-diesel vessel (HPDV) and HFO vessel (HFOV) (see Table S1 for detail). The parameters of all the fuels used for each tested vessel are shown in Table S2. A

total of 28 sets of size-segregated samples were obtained, shown in Table S3. The main operating modes of each vessel were chosen according to actual operating conditions. Information about the sampling systems and fuels used for the tested vessels can be found in our previous studies (Zhang et al., 2016; Zhang et al., 2018).

### 2.2 Sampling instrument

An 8-Stage Andersen Cascade Impactor (TE-20-800, Tisch Environmental Inc, USA) was used for particle collection. The flow rate was 28.3 liter per minute. The particles were separated into 9 size bins by the impactor, and the particle size range, sampling spot number/sampling area for each stage are shown in Table S4. The sampling duration for each sample varied from 10 to 30 minutes according to

emission conditions and dilution ratios.





### 2.3 Chemical analysis

The mass, organic carbon, element carbon, sixteen priority PAHs indicated by US EPA (the detailed information is shown in Table S5), and n-alkanes from C10 to C34 in each particle size bin were measured. The mass of the particles on each filter was obtained by gravimetric method. OC and EC were measured with a 0.544 cm$^2$ punch aliquot of each filter sample by thermal optical reflectance (TOR) following the IMPROVE-A protocol with a DRI Model 2001 Thermal/Optical Carbon Analyzer (Atmoslytic Inc., Calabasas, CA). The measuring range of TOR was from 0.05 to 750 μg C cm$^{-2}$ with an error of less than 10%. The PAHs and n-alkanes were measured by an optical-4 thermal desorption (TD) sample injection port coupled with an Agilent GC7890B/MS5977A (Agilent Technologies, Santa Clara, CA) system (Han et al., 2018). The detailed information of TD-GC/MS method for PAHs and n-alkanes was shown as follows.

Each of the collected Quartz filter samples was punched with an area of 5 mm in diameter with a sampling spot as the center for PAHs and n-alkanes analysis. The filter was cut to small pieces and loaded into a TD glass tube. 1 μL deuterium marked compounds and 6-methyl benzene with concentrations of 10 ppm was injected into the tube as internal standards. The TD glass tube was then placed into the TD inject port, and was heated to 310 ℃ at a rate of 12 ℃/min and thermally desorbed at 310 ℃ for 3 min. The desorbed organic compounds were trapped on the head of a GC-column (DB-5MS: 5% diphenyl-95% dimethyl siloxane copolymer stationary phase, 0.25 mm i.d., 30 m length, and 0.25 mm thickness). The initial GC oven temperature was 60 ℃ and held for 4 min, then rose to 300 ℃ at a rate of 6 ℃/min and was held at 300 ℃ for 8 min (Han et al., 2018). D8-Naphthalene, D10-acenaphthene, D10-phenanthrene, D12-chrysene, D12-perylene, D-C16, D-C20, D-C24, and D-C30 were used for the analytical recovery check. The detection limit for the TD-GC/MS method ranged from 0.2 pg mm$^{-2}$ (Ace) to 0.6 pg mm$^{-2}$ (Icdp).

### 2.4. Quality assurance/quality control

Rigorous quality assurance and control were conducted during the whole





experiments. Filter blanks were analyzed in the same way as the above procedures to
determine the background concentration. Duplicate samples as well as standard
samples were examined after analyzing a batch of 10 samples to ensure that the error
was within 5%. The average recoveries of the deuterium surrogates ranged from 84.3%
to 101% in this study (shown in Table S6). The results of each sample were subtracted

by the filter blank results. The final data reported in this study were not corrected by
the recoveries.

### 3 Results and discussions

### 3.1 Total mass distributions in different particle size bins

Emission factors for the total PM and size-segregated particle mass distributions

of all the 12 tested ships with different modes are shown in Fig. 1. The emission factor
for PM was discussed in our previous studies (Zhang et al., 2016; Zhang et al., 2018;
Zhang et al., 2019). In general, low engine power fishing boats had much higher PM
emission factors than other types of ships, while high engine power diesel vessels
showed lower PM emission factors, especially for the high quality fuel ships. Besides,

low load operating modes showed higher PM emission factors for almost all the tested
ships. In this study, we focused on the size-segregated particle mass distributions of
the tested ships. It was found that more than half of the particle mass was
concentrated in fine particles with $D_p<1.1$ μm (33-91%). However, there were still
large proportions (5-53%) in coarse particles with Dp>3.3 μm. The findings were in

line with the previous studies which reported that the mass distributions of ship
emissions were dominated by accumulation mode and/or coarse mode particles
(Moldanová et al., 2009, 2013; Murphy et al., 2009). In contrast, the total particle
number concentrations were dominated by nucleation mode particles with $D_p<0.1$ μm,
while particles with $D_p>0.5$ μm showed very low number concentrations that were

often neglected in previous studies (Cappa et al., 2014; Beecken et al., 2014;
Moldanová et al., 2013; Juwono et al., 2013; Diesch et al., 2013; Alfoeldy et al., 2013;
Winnes and Fridell, 2010; Kasper et al., 2007).



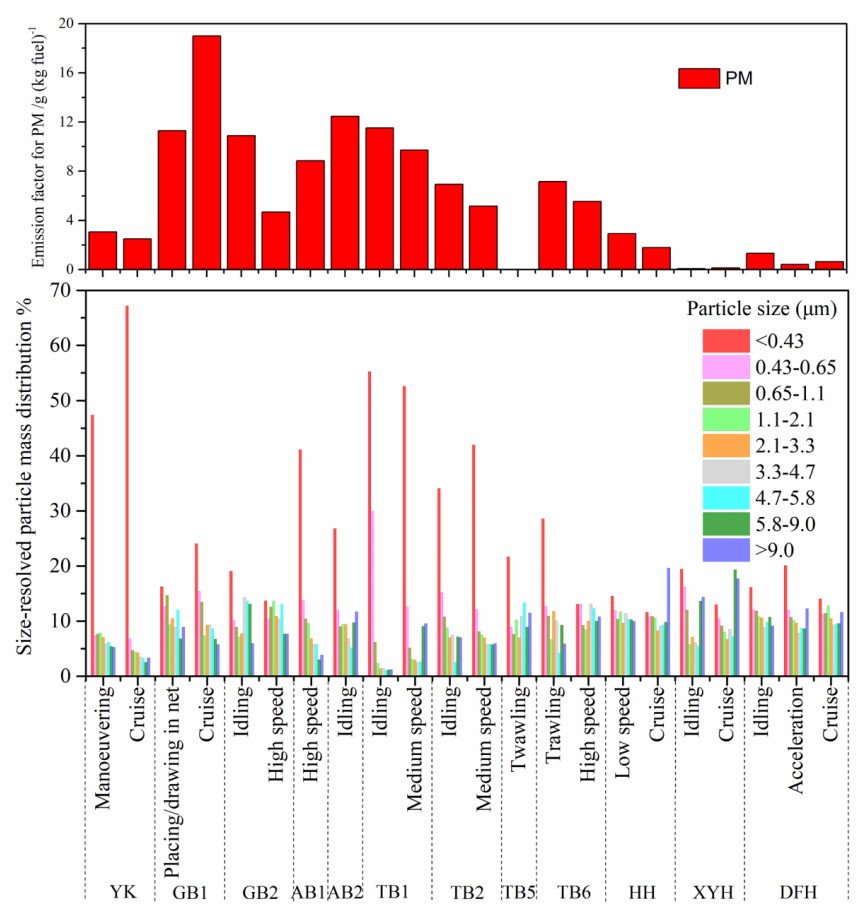

Figure 1 Emission factor for total PM and size-resolved particle mass distributions
with different modes for the 12 tested ships

Different types of ships showed rather different size-segregated particle mass distributions. As shown in Fig. 1, the mass was concentrated on particles with $D_p < 0.43$ μm for the HFO driven ship YK, coincident with the results of a previous study that found the particle mass peaked at around 50 nm for a HFO ship (Chu-Van et al., 2017), but inconsistent with a study on hot-exhaust particles from a HFO driven cargo vessel which revealed bimodal particle mass distribution with peaks at ~ 7 μm and ~ 0.5 μm, attributed to the HFO combustion and the various nature and composition of the ship-exhaust particulates (Moldanová et al., 2009). In this study



we found that high power diesel vessels had relatively smaller fractions of particle

mass in the fine mode with $D_p$<0.43 μm but larger fractions in the coarse mode with

$D_p$>5.8 μm than those of the HFO ship. This pattern was also consistent with the

result of Fridell et al. (2008) who found a large peak at ~10 μm from the emissions of

large diesel engines. Furthermore, different fishing boats (GB1 to TB6 in Fig 1.)

showed different size-segregated particle mass distributions in this study. It can be

seen that low engine power fishing boats had high percentages of fine particles with

$D_p$<1.1 μm, while high engine power fishing boats showed high proportions of coarse

particles. The different size-segregated particle mass distributions for different types

of ships were likely caused by the quality of fuel and its combustion efficiency in the

engines. For example, compared to diesel fuel, HFO contains high percentage of

aromatics, which are known to contribute to nucleation mode particle formation

(Zetterdahl et al., 2017). Moreover, incomplete combustion could enhance the

formation of nucleation mode particles from the unburned fuel and lubrication oil

(Zetterdahl et al., 2017). Hence, the relatively lower combustion efficiency of HFO

ship led to higher proportions of fine particles than diesel-fueled ship. Moreover, the

quality of fuel used for the same type of ships such as fishing boats varied largely in

China, which might, at least to some extent, was also responsible for the different size

distributions. Compared to high engine power fishing boats, relatively lower

combustion efficiencies of the low engine power fishing boats resulted in higher

proportions of fine particles (Zhang et al., 2018).

The size-segregated particle mass distributions in different operating modes were

also compared in this study. No obvious discrepancy was found for all the tested ships,

similar to a previous study on mass distribution from measurements onboard of three

ships (Fridell et al., 2008), but different from another study on a marine engine that

reported the particle mass distribution centered at 0.1-0.2 μm with much fewer coarse

particles under at-berth condition compared to maneuvering and ocean-going

conditions (Chu-Van et al., 2017). The main reason was that the at-berth emission was

calculated based on auxiliary engine but not the main engine by Chu-Van et al. (2017).



Coincidently, emissions from two auxiliary engines in the HFO driven ship YK and
engineering ship HH were measured in this study (Fig. S1). Both showed high
proportions of fine particles with $D_p$<0.43 μm and small percentages of coarse
particles, similar to the findings of Chu-Van et al. (2017). Overall, fuel type, fuel
quality, engine type might have higher influence on particle mass distributions from
ships than the operating mode.

**3.2 Characteristics of OC and EC in size-segregated particles**

**3.2.1 The OC and EC mass distributions in different particle size bins**

Figure 2 and Table S7 present the total OC and EC mass distributions in different
particle size bins for the three ship types. The OC and EC distributions in different
particle size bins showed similar but distinguished characteristics compared to the PM
mass distribution. All the tested ships presented similar trends, namely, the
proportions of OC and EC decreased with the increase of particle size. About 53-86%
of OC and 68-86% of EC were in the particles with $D_p$<1.1 μm. Only very small
percentages of OC and EC existed in the coarse particles. Among the three types of
ships, HFOV showed significantly higher OC and EC proportions in particles with
$D_p$<0.43 μm (75% and 66%, respectively) than the other two types of ships operated
with diesel fuel, while the OC and EC proportions were lower in the other particle
size bins of HFOV vessels than those of the other two types of ships. HPDV ships
only accounted for 23% OC and 27% EC in particles with $D_p$ < 0.43 μm, which were
in accordance with the characteristics of total PM mass distributions; that is, diesel
vessels had relatively smaller proportions of fine particles with $D_p$<0.43 μm and larger
proportions of coarse mode particles than HFO ships. Notably, the proportion of EC in
particle size bin of 0.43-0.65 μm had almost the same level as that in particle size bin
of less than 0.43 μm for LPDF. The higher proportions of OC and EC in fine particles
from the HFO vessel compared to the diesel fuel ships might be attributed to both the
combustion efficiency and the use of HFO. Due to the large fuel to air ratio in the
HFO engine (Kittelson et al., 1998), higher fractions of unburned fuel with heavier
molecular weight as well as lubrication oil could more easily leak into the exhaust and



raise the carbonaceous fractions (Hardy and Reitz, 2006). Besides, the incomplete

combustion and the high content of aromatics in HFO both could enhance the

formation of new nucleation mode particles (Zetterdahl et al., 2017), which would

eventually promote a high fraction of fine mode carbonaceous particles.

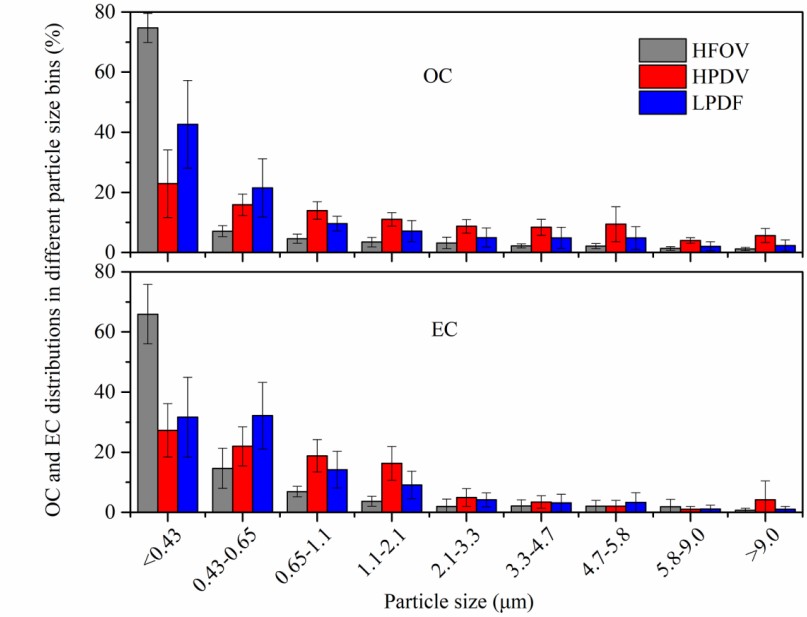

Figure 2 Total OC and EC distributions in different particle size bins

### 3.2.2 OC and EC proportions in each particle size bin

In order to figure out the carbonaceous components in size-segregated particles,

percentages of OC and EC in each particle size bin are given in Fig. 3. In general,

both OC and EC proportions showed overall decreasing trends with the increase of

particle sizes for all three types of ships. The OC+EC accounted for large proportions

of the total particle mass in fine particles, between 40 and 65 % for particles smaller

than 0.65 μm. However, OC+EC only explained small proportions in coarse particles,

less than 15% for particles larger than 5.8 μm diameter, suggesting that most of the

coarse particle mass was dominated by other non-carbonous components, such as ash

and hydrated sulfates. The coarse particles could be seriously influenced by

natural-based substances such as ash that was introduced into the cylinder by the air to



maintain a certain-reliable equivalent ratio in the cylinder. This was confirmed in a previous study about the ship-exhaust particle composition based on transmission electron microscopy (TEM) study. Mineral/ash particles containing lime, calcite, vanadium oxide and nickel sulfide were also found to be dominant in coarse particles

(Moldanová et al., 2009).

In addition, the differences of OC+EC proportion patterns among the three types of ships were large. OC+EC accounted for >60% in particles with $D_p$<0.65 μm for HFOV, then decreased sharply to less than 30% in particles with $D_p$>0.65 μm, and only explained ~10% in particles with $D_p$>9.0 μm. The OC+EC for LPDF showed

similar trend to that for HFOV, but had higher proportions in particles with $D_p$ of 0.65-3.3 μm. In contrast, the OC+EC only accounted for ~40% in particles with $D_p$<1.1 μm for HPDV, and decreased to less than 10% in particles with $D_p$>9.0 μm. Because the OC and EC compositions in size-segregated particles from ships were not studied in the past, this study compared components of $PM_1$ or hydrated sulfate and

organic carbon in size-resolved particles with earlier studies. Diesch et al. (2013) found that organic matter (OM) was the most abundant component in $PM_1$, while EC contributed only 6% on average to $PM_1$. This result was similar to the composition of HFOV emissions in this study but significantly different from that of the HPDV and LPDF, likely due to the different types of engine and fuel. Moreover, Healy et al.

(2009) found that hydrated sulfate and organic carbon were the dominant components of size-resolved PM from ship plumes, while Murphy et al. (2009) reported that the fraction of unknown mass was much higher at small particle sizes, which were different from our results. The large discrepancies might be caused by the different sampling, analytical methods and the fuel quality. In addition, the unidentified

components in each particle size bin should be clarified in the future, especially in coarse particles.



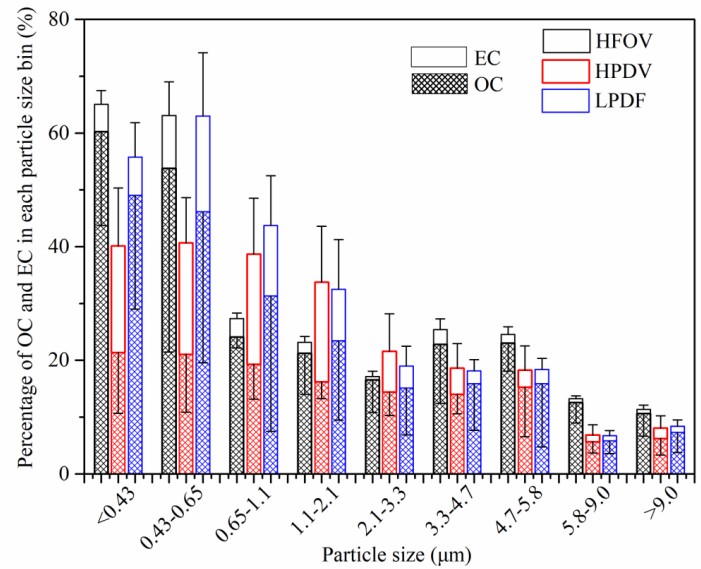

Figure 3 Percentage of OC and EC mass in each particle size bin

The average proportions of OC were higher than EC in each individual particle size bin for all the tested ships. For better comparison, OC to EC ratios in all particle size bins are given for the three types of ships in Fig. 4. The OC/EC ratio of HFOV in each particle size bin was the highest, followed by LPDF and HPDV. The OC/EC ratios in fine particles from HFOV were more than 10, while they were less than 3 for

HDPV. In addition, with the increase of particle size, the OC to EC ratios decreased first and then increased, with the lowest values in particle sizes of 0.43-1.1μm. However, all the ships had similar median OC/EC ratios of ~10 in coarse particles with $D_p$>4.7 μm.





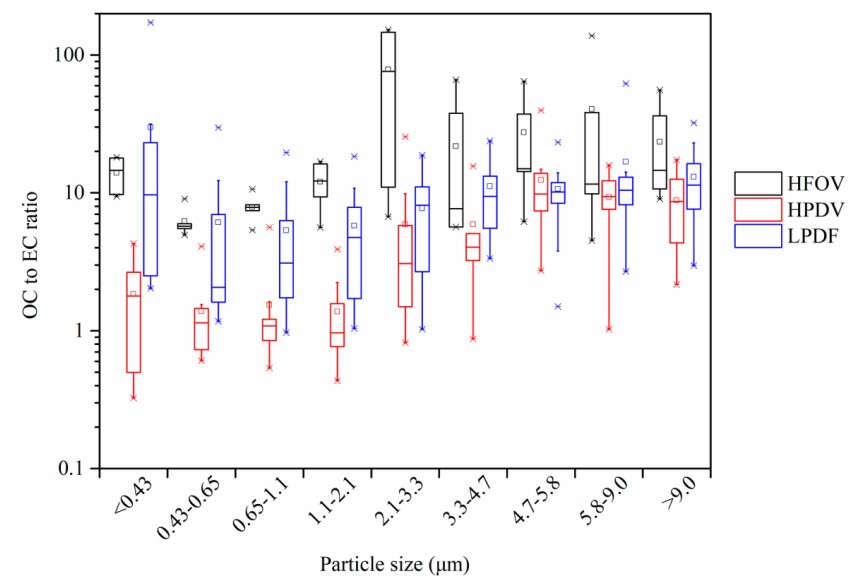

Figure 4 OC to EC ratio in different particle size bins

It is well known that particles emitted from diesel engines are formed through a complex process, starting with soot precursors generated from thermal decomposition of the large hydrocarbon molecules, through processes such as dehydrogenation, PAH
formation, growth in molecular weight, particle inception, growth by surface reaction and coagulation, agglomeration and oxidation, etc. (Lighty et al., 2000). In the processes, several factors can influence the change of particle number (PN) and diameter: coagulation reduces the PN through particle collision that forms new particle; surface growth involves the attachment of gas-phase species to the existing
particles and results in no changes in PN but increase of PM diameter and mass; aggregation leads to chains and clusters of primary soot particles and an increase in particle size (to 10-100 nm in diameter) (Heywood, 1988); and oxidation occurs during the formation and growth processes of the particles, resulting in the formation of gaseous species and reduction of soot particles and their precursors. In addition, the
exhaust gases continue to cool down and are diluted with air in the exhaust gas channel. The decreased temperature has an impact on the condensation and adsorption



of hydrocarbons and other volatile compounds, but not on the soot particles (Heywood, 1988). The volatile compounds in the exhaust gas channel may form new particles via transformation, nucleation, adsorption and condensation, depending on the conditions in the exhaust gas channel (Kittelson, 1998).

Soot particles were mainly formed through pyrolysis of diesel fuel and lubrication oil, and the organic fraction was formed through incomplete combustion of fuel and lubrication oil (Zetterdahl, 2016). Incomplete combustion of fuel and lubrication oil could significantly enhance the formation of fine particles, which might be the reason for the higher OC to EC ratio in particles with Dp<0.43 um. This study found that most EC were in the particles with Dp<1.1 um, coincident with the fact that soot was primarily in accumulation mode with particles of 0.1<Dp<1 um (Kasper et al., 2007). Although OC was also concentrated in fine particles with $D_p$<1.1 μm (Section 3.1), its percentage was higher in particles with $D_p$<0.43 μm, and lower in particles with 0.43<$D_p$<1.1 μm, compared to the EC distribution. Therefore, OC to EC ratios were lowest in particles with 0.43<$D_p$<1.1 μm for all the tested ships. For the coarse particles, though non-carbonous components were the dominant parts, OM increased after the cooling of the exhaust gas while content of EC and ash remained unchanged through adsorption of hydrocarbons and other volatile compounds (Zetterdahl, 2016), which enhanced the formation of coarse particles. Consequently, OC to EC ratios also showed high values in coarse particles with $D_p$>3.3μm. HFOV showed higher OC to EC ratio than the diesel ships, which might be caused by the relatively lower combustion efficiency. Similarly, because HPDV ships had higher combustion efficiencies due to the higher fuel quality and better engine maintenance in this study, OC to EC ratios of HPDV showed the lowest levels.

### 3.2.3 OC and EC fragments in size-segregated particles

According to the IMPROVE-A protocol used in the thermal-optical carbon analysis, fragments of OC (OC1,OC2, OC3, and OC4 fragments obtained in 120, 250, 450, and 550 ℃ in pure He atmosphere, respectively) and EC (EC1, EC2, and EC3 fragments obtained in 550, 700, and 840 ℃ in 98% He/2% O$_2$ atmosphere,





respectively) were obtained and used to understand different formation processes (Sippula et al., 2014). In this study, percentages of OC and EC fragments in different particle size bins are shown in Fig. 5. Typically, OC1+OC2 were classified as volatile organic compounds while OC3+OC4 were categorized as refractory organic compounds. EC was divided into char and soot. EC1 was classified as char-EC and EC2+EC3 were classified as soot-EC (Han et al., 2018). OC1, OC2 and OC3 were the dominant OC fragments for all the tested ships, while EC2 was the prevailing EC fragment for diesel fuel ships and EC1 was the main EC fragment for HFO ship. In comparison, OC1+OC2 in LPDF ships accounted for higher percentages than other types of ships, especially OC1, while OC3+OC4 in HFOV had more fractions than in LPDF and HPDV, revealing that more pyrolytic organic matters were emitted from HFOV ships.

The results are consistent with a previous study which found that 37-57% of the OC were heavier OC fragments (OC3, OC4, and pyrolytic carbon) in PM emitted from heavy fuel oil ships, while the PM from diesel fuel ships was dominated by the most volatile OC1 and OC2 fragments (Sippula et al., 2014). Since OC was formed through incomplete combustion of the fuel and lubrication oil, the different OC fragments in PM between HFO and diesel ships (LPDF and HPDV) reflected the variations of the fuel composition. This result was coincident with the fact that heavy fuel oil has longer carbon chains than diesel fuel. For EC, the amount of soot particles emitted from cylinder depends on the difference between the formation rate and the oxidation rate during the expansion stroke (Zetterdahl, 2016). From Fig. 5 we inferred that the 2-stroke low speed HFO engine (HFOV) had higher oxidation rate than the 4-stroke medium/high diesel engines (HPDV and LPDF), leading to lower soot-EC percentages.

The OC and EC fragments had different percentages in different size particles. On one hand, OC1 in diesel fuel ships presented obvious decreasing trend with the increase of particle size. However, though the highest proportion of OC1 was observed in particles with $D_p<0.43$ μm for HFOV, there was an opposite variation





trend of the other particle size bins to that of diesel fuel ships. Furthermore, for diesel
fuel ships, both OC2 and OC3 had lower proportions in small particles and higher
proportions in coarse particles. For HFOV, OC2 accounted for the largest proportion
in particles with $D_p<0.43$ μm, whereas its proportion in other particle size bins was
similar. In contrast, OC3 occupied the smallest proportion of HFOV in particles with

$D_p<0.43$ μm, and was constant in other particle size bins. In addition, OC4 showed no
significant variations among the particle size bins for all the tested ships. On the other
hand, EC1 showed decreasing trends with the increase of particle size for all the tested
ships, while EC2 had the highest proportions in particles with $0.65<D_p<1.1$ μm for
diesel fuel ships, with overall higher proportions in small particles and lower

proportions in coarse particles. However, EC2 of HFOV showed no significant
variations. In addition, EC3 had very small proportions for all the tested ships and
showed no significant variations with particle sizes.

It is known that particles in the nucleation mode consist of both solid particles
and condensable organic and sulphur compounds that are usually formed during

dilution and cooling of the exhaust gas (Kittelson, 1998). Hydrocarbons and carbon
fragments are the main source of tiny particles from engines running under normal
conditions with low-sulfur fuels (Kittelson, 1998). Besides, it has been proved that
new particle formation via nucleation is more favorable than adsorption on exiting
particles if there is small surface area on which to adsorb when hydrocarbons transfer

from gas phase to particle phase (Kittelson, 1998). Therefore, high volatile organic
matter in nucleation mode particles might be the reason for higher proportions of OC1
in small particles. It is also known that the majority of accumulation-mode particles
are solid agglomerates with adsorbed compounds (Kittelson, 1998). Due to the
smaller surface areas of accumulation and coarse mode particles, fewer volatile

organic matters can be adsorbed with the increase of particle size, leading to the
decreased proportions of OC1. In addition, OC3 and OC4 stem from low volatile
organic matters with large molecular weights, which might be formed directly inside
the cylinder under high temperature and pressure. The existence and formation of





large size particles containing OC3 and OC4 have been proved in a previous study (Han et al., 2018). In this study, EC was primarily found in the accumulation mode. This is to some extent consistent with the findings of Kittelson (1998) that accumulation mode particles are mainly carbonaceous soot agglomerates formed directly by combustion, and those of Moldanová et al. (2009) that char and char-mineral particles were concentrated in a size range of 0.2-5 μm. Moreover, the

decreased temperature in the exhaust gas channel has an impact on the condensation and adsorption of hydrocarbons and other volatile compounds during the coarse particle formation, but not on the soot particles (Heywood, 1988). As a result, EC fragments showed obvious deceasing trends with the increase of particle size in coarse-mode. The formation of OC and EC fragments could be influenced by many

factors, such as engine condition (temperature, pressure), fuel type (fuel composition and structure), operating conditions, etc. (Tree and Svensson, 2007). However, detailed formation mechanism of OC and EC fragments in size-segregated particles still need to be further studied.

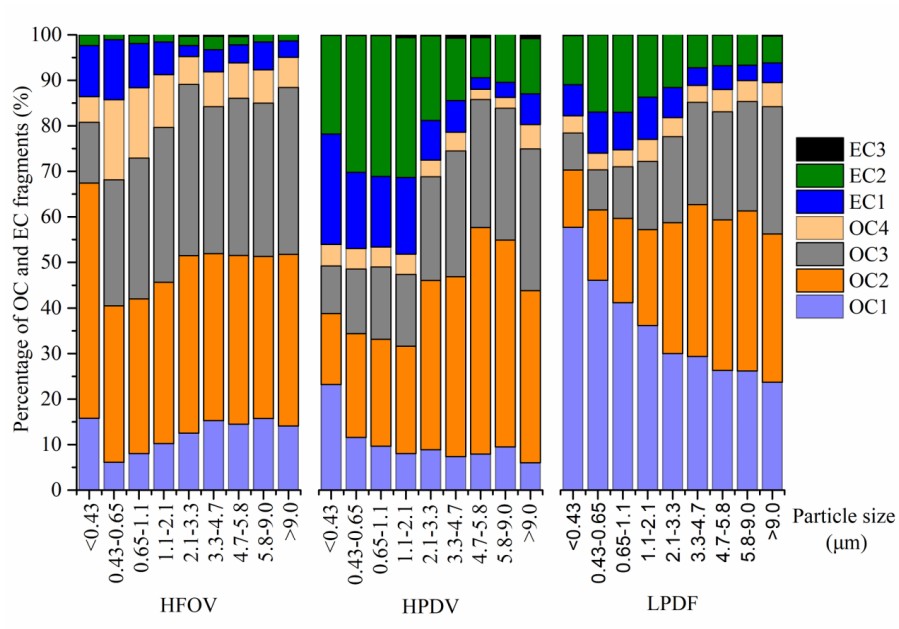

Figure 5 Percentage of OC and EC fragments in different particle size bins


### 3.3 Characteristics of PAHs and n-alkanes in size-segregated particles

### 3.3.1 Distributions of total PAHs and total n-alkanes in different particle size bins

The distributions of total PAHs and total n-alkanes in different size particles emitted from these three types of ships in China are shown in Fig. 6 and Table S8. Clearly, large proportions of identified organic compounds were concentrated in fine particles. About 66.3% to 88.0% of PAHs were in particles with $D_p < 1.1$ μm. The results are consistent with the finding of a previous study that revealed >69% particle-PAH was associated with $PM_{2.5}$ emitted from residential coal combustion (Shen et al., 2010). Similar observations were also reported from indoor crop burning (Shen et al., 2011) and motorcycles (Yang et al., 2005). In addition, a previous study about PM from a typical container ship has demonstrated that the smaller the particles are, the greater their toxicity is (Wu et al., 2018). In comparison, n-alkanes in fine particles with $D_p < 1.1$ μm accounted for higher proportions than PAHs, with a range of 79.0~94.6%. Both PAHs and n-alkanes in particles with $D_p < 1.1$ μm from HFOV had the highest proportions among the three types of ships, and they were the lowest from HPDV. Thus, large proportions of PAHs (33.7%) and n-alkanes (21.0%) were still in particles with $D_p > 1$ μm for HPDV, which were seldom reported in previous studies of ship engine exhausts.

It is well documented that incomplete combustion of fuel and lubrication oil was one of the major sources of organic compounds in PM and enhanced the new particle formation (Agrawal et al., 2010; Zetterdahl et al., 2017). Higher PAHs contents in heavy fuel oil and lower combustion efficiencies in 2-stroke lower speed engines could lead to enhanced nucleation-mode particle formation, which could contain more PAHs and n-alkanes. In contrary, lower PAHs contents in diesel fuels of HPDV ships, together with better maintenance of the engines, might lead to better combustion conditions and less nucleation-mode particle formation that had lower organic matter proportions. In addition, new particles could be formed in the exhaust gas channel through transformation of nucleation, adsorption and/or condensation. Therefore, the varied conditions in the exhaust gas channels caused the differences in organic matter



proportions in coarse particles. The relatively longer exhaust gas channel and higher temperature in the channel of HFOV enhanced the formation of new secondary nucleation-mode and/or accumulation-mode particles from gas-phase organic compounds, leading to higher proportions of organic matters in fine particles

(Kittelson, 1998).

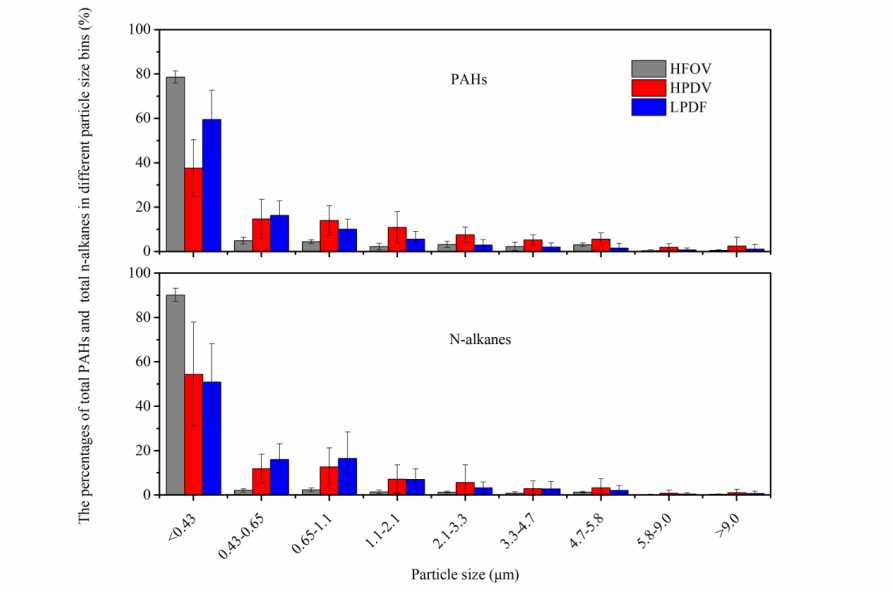

Figure 6 The distribution of total PAHs and total n-alkanes in different particle size

bins (%)

**3.3.2 Emission ratios of PAHs and n-alkanes in each particle size bin**

In order to explore the variations of total PAHs and total n-alkanes in different size particle bins, the emission ratio ($\mu$g (g$^{-1}$ PM) and mg (g$^{-1}$ PM)) was defined and discussed in this study. It is calculated by dividing the total PAHs or n-alkanes mass ($\mu$g/mg) through the particle mass (g) in each particle size bin. The emission ratios of

total PAHs and total n-alkanes in size-segregated particles are shown in Fig. 7. Obviously, the emission ratios of both PAHs and n-alkanes decreased with the increase of particle size, and they were more than two orders of magnitude higher in fine particles than in coarse particles. Namely, much more organic matter was





contained in fine particles than in coarse particles. This is consistent with previous

findings that nucleation-mode particles usually contain a large proportion of organic

compounds (Kittelson, 1998; Moldanováet al., 2009).

Among the three types of ships, LPDF had the highest PAHs emission ratios in

fine particles with $D_p$<1.1 μm, and the HFOV had the lowest values. However, no

significant difference of PAHs emission ratios was observed in particles with $D_p$>1.1

μm for all the tested ships ($p$>0.05), so was for n-alkanes. In particular, the low power

diesel fishing boats in China had not only high levels of PM emission factor, but also

high proportion and emission ratio of organic matters in fine particles, implying their

severe influence on human health and the environment.

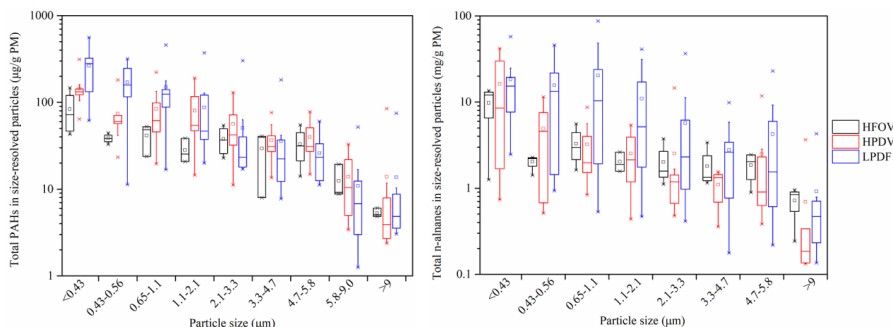

Figure 7 Emission ratiosof total PAHs (in μg (g$^{-1}$ PM)) and n-alkanes (in mg (g$^{-1}$ PM))

in size-resolved particles

### 3.3.3 Profiles of speciated PAHs and n-alkanes in size-segregated particles

The speciated profiles of PAHs and n-alkanes in size-segregated particles emitted

from the three types of ships are shown in Fig. 8. Phe, Flua, Pyr, BaA, Flu and Chr

(full names and their abbreviations are shown in Table S5) were the most dominant

PAHs, while their relative percentages in different types of ships and in different

particle size bins varied. For HFOV, the levels of Flua, Phe, BaA and Pyr were higher

than other PAH species, while the profiles of PAHs in different particle sizes were

similar. For HPDV, Phe, Flu, BaA and Pyr were the main PAHs, with Phe having the

highest proportions in all the particle size bins. Furthermore, the percentages of Phe

and BaA were lower in coarse particles. This is in accordance with the fact that



synthetic PAHs have higher concentrations in freshly formed particles (Pergal et al.,
2013). However, the percentages of Pyr and Chr increased with the increase of

particle size. For LPDF, Pyr, BaA, Phe, Flua and Chr were the dominant components.
The percentage of BaA decreased with the increase of particle size, whereas Pyr and
Chr presented opposite trends. The various profiles of PAHs in different particle size
bins indicated their different formation processes, which were likely related to the fuel
quality, engine type and the condition in the exhaust gas channel (Lombaert et al.,

2006). It should be noted that when diagnostic characteristics of PAHs in PM emitted
from ships is used in source apportionment, particle size needs to be taken into
consideration.



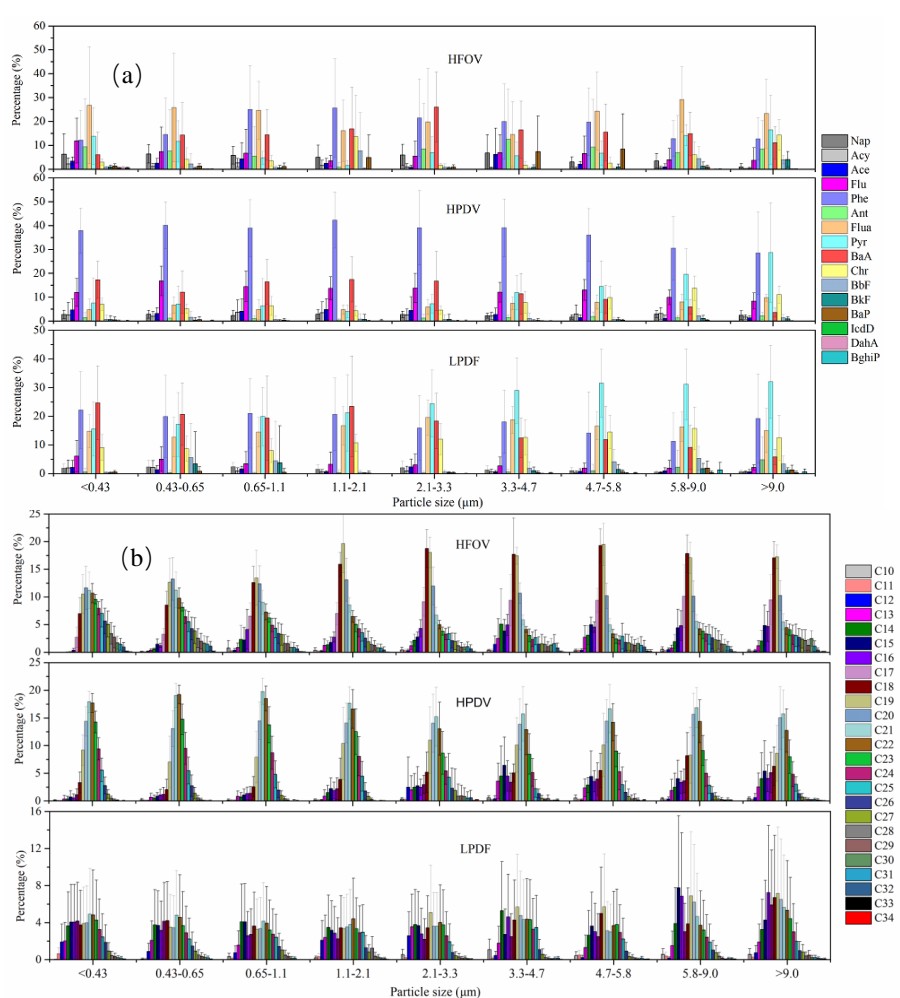

Figure 8 Speciated profiles of PAHs (a) and n-alkanes (b) in size-resolved particles


The ratio of high molecular weight (HMW) PAH (four to six rings) to low molecular weight (LMW) PAH (two and three rings) is shown in Fig. S2. The LPDF had the highest HMW to LMW ratios (2.96-8.25), while the HPDV had the lowest ratios (0.62-1.98). The results indicated that LPDF and HFOV had unfavorable

combustion conditions compared to HPDV, which is consistent with our previous conclusion. The HMW to LMW ratios were generally higher in fine particles and decreased with the increase of particle size, except for HFOV which still showed high



ratios when particle size was larger than 5.8 μm. Since HMW PAHs have high toxic equivalence (TEQ), the higher HMW/LMW PAHs ratios in fine particles implied that small particles emitted from the three types of ships might be more toxic than coarse particles. Furthermore, from our previous study the LPDF showed not only higher HMW to LMW ratios than other ships, but also much higher emission amount of total PAHs (Zhang et al., 2019), implying the necessity of more stringent control on the emissions of particles from fishing boats in China.

From Fig. 8, it can be seen that n-alkanes showed different percentage distributions in different particle size bins from different types of ships. There were larger maximum carbon numbers ($C_{max}$) in smaller size particles, such as C20/C21 for HFOV, C21/C22 for HPDV and C21/C22 for LPDF. However, with the increase of particle size, the $C_{max}$ of n-alkanes switched to C18/C19, C20/C21 and C18/C19 for HFOV, HPDV and LPDF, respectively. Similar to PAHs, when diagnostic characteristics of n-alkanes in PM emitted from ships is used in source apportionment, particle size needs to be taken into consideration.

## 4. Conclusions and implications

Particulate matters emitted from ships have gained more and more attention in recent years due to their adverse impacts on human health and ambient air quality. However, our knowledge on the abundance and chemical speciation in different particle sizes are limited. In this study, size-segregated distributions and characteristics of mass, OC, EC, 16 PAHs and 25 n-alkanes from 12 different vessels in China were presented.

This study found that over 50% of the total particle mass was explained by the particles with $D_p$<1.1 μm for most of the tested ships. Specifically, the mass was concentrated in particles with $D_p$<0.43 μm for HFOV, and in particles with $D_p$<1.1 μm for low power diesel fishing boats, while the mass was accounted for higher percentages by coarse particles for high power diesel fishing boats and HPDV.

Similar to the total particle mass distribution, about 53-86% of total OC and 68-86% of total EC were in the particles with $D_p$<1.1 μm, presenting downward trend





with the increase of particle size. The OC+EC accounted for major proportions of the total particle mass in fine particles. However, OC+EC only explained small proportions in coarse particles, suggesting that most of the coarse particle mass was

dominated by other non-carbonous components. In addition, the OC to EC ratio of HFOV in each particle size bin was the highest, followed by LPDF and HPDV. With the increase of particle size, the OC to EC ratios decreased first and then increased, with the lowest values in particle sizes of 0.43-1.1μm. Moreover, fragments of OC and EC were obtained and used to understand different formation processes. OC1,

OC2 and OC3 were the dominant OC fragments for all the tested ships, while EC2 was the prevailing EC fragment for diesel fuel ships and EC1 was the main EC fragment for HFO ship. Different OC and EC fractions showed different distributions in different particle size bins because of the different formation mechanisms.

     16 PAHs and 25 n-alkanes in different particle bins were identified in this study.

Results showed that large proportions of the organic compounds were concentrated in fine particles, with about 66.3 - 88.0% of PAHs and 79.0 - 94.6% of n-alkanes were in particles with $D_p$<1.1 μm. LPDF had the highest PAHs and n-alkanes emission ratios in fine particles with $D_p$<1.1 μm, and the HFOV had the lowest values, but no significant differences were observed in particles with $D_p$>1.1 μm. Results of the

speciated profiles of PAHs showed that Phe, Flua, Pyr, BaA, Flu and Chr were the most dominant PAHs, while their relative percentages in different types of ships and in different particle size bins varied. In addition, the HMW PAHs to LMW PAHs ratios were generally higher in fine particles and decreased with the increase of particle size. N-Alkanes also showed different percentage distributions in different

particle size bins from different types of ships. Moreover, there were larger $C_{max}$ in smaller size particles, but switched to smaller $C_{max}$ with the increase of particle size for all the tested ships.

     This study confirmed that the particle mass distributions in different size bins from ships were significantly different from the particle number distributions, and

different types of ships had their own distinct mass distributions. The different profiles



of chemical components in size-segregated particles implied that size-segregated chemical profiles should be considered when source apportionment was conducted. Furthermore, this study found more toxic organics such as PAHs in small particles emitted from fishing boats, suggesting the necessity of more stringent control on this

type of boats in China. In addition, detailed formation mechanisms of chemical composition in size-segregated particles need in-depth investigation. Finally, the large proportion of unidentified components in coarse particles still needs to be figured out. Due to the limitation of instrumental analysis, only very small proportion of specific organic compounds in each size particle bin were identified in this study, implying

that innovative analytical technology such as real-time measurements of particle-associated organic compounds using thermo-desorption aerosol gas-chromatograph approach should be urgently developed.

**Data availability.** The data used for this study can be obtained from fzhangtj@tongji.edu.cn upon request.

**Author contributions.** FZ did the sampling, led the writing and data analysis for the manuscript with significant contributions and comments from all co-authors. YC designed the field measurements and HG reviewed and edited the manuscript. VM, YZ, XY and JC contributed to the data analysis, the manuscript structure, and the writing of the text.

**Competing interests.** The authors declare that they have no conflict of interest.

**Acknowledgement.** The authors wish to thank Min Cui, Yong Han, Zhe Qian, Yajing Lu for the help of experimental work. This study was financially supported by the Natural Science Foundation of China (grant numbers 41761134083, 91744203, 41603090, and 21677038), National Research Program for Key Issues in Air Pollution

Control (grant numbers DQGG0201), and the Collaborative Research program between the Beijing University of Technology and the Hong Kong Polytechnic University (PolyU) (4-ZZFW).

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
