# Peer review of "Size-segregated characteristics of OC, EC and organic matters in PM emitted from different types of ships in China"

_Atmospheric Chemistry and Physics, 2019_

## Referee Comment (RC1) · Anonymous Referee #2 · 2 Sep 2019

The issue discussed in this manuscript is important. If authors could satisfy reviewers and readers on the following major questions, this paper could be considered by ACP. 1. It's authors' duty to explain what's new in this paper compared to several other papers published by the same group, especially the one on Atmospheric Environment 2019. Both titles indicate similar contents. 2. The results were not organized in a clear way which made them very hard to follow. 1) For different particle size bins, what are the chemical component profile? No figure gives a comprehensive component profile. Only the OC/EC percentages compared among bins were provided. Authors failed to touch the whole picture of the "size-segregated characteristics", which should be the most important part of this study. For example, EC and OC were found very

low for coarse PM. Then, which components are the major part for coarse PM? In addition, without a total mass analysis, it's impossible to judge the reliability of the sampling and analysis. 2) If the whole picture of size based chemical profile could be provided, then it's OK to discuss the distribution for each component in different size bins. However, this information is not very important compared with part 1). Currently, authors spent too much pages on discussing this, including Fig. 2, 6 and etc. 3) All figures were displayed in percentage or ratio. The mass of OC, EC, PAH or others should be provided directly. Is it still necessary to provide so many figures if the mass could be given? 4) Figure 1 is confusing. 'YK, GB1...' should be replaced with ship types, e.g. 'HDPV'.

3. Presentation quality in text also needs to be improved. 1) In abstract, line 30, 'in fine particles, OC and EC were the dominant components'. line 34, 'OC and EC have the lowest values for 0.43 to 1.1 um'. Are they still dominant? 2)Line 34, What are the OC1, OC2 and OC3? 3) Line 214, how can 5% be called the large proportion? 4) line 282, HFOV vessels should be HFOV. and 'HPDV ships' should be 'HPDV'. And this sentence is confusing. What's the meaning by 'HPDV accounted for 23%....'? Compared with what kind of ships?
* * *

---

## Referee Comment (RC2) · Anonymous Referee #1 · 18 Sep 2019

General comments:

Results are interesting and new and the language is overall good.

I lack a discussion on the relevance of the study in a larger context. It should be elaborated upon from the viewpoint that the engines are relatively small, high speed marine engines, and only representative for smaller marine craft.

I find that very little attention is given to the engine and combustion characteristics. It is stated that "Overall, fuel type, fuel quality, engine type might have higher influence on particle mass distributions from ships than the operating mode", but it seems from Figure 1 that also operational mode is important for individual ships. Results on EC and

[Figure]

OC at different engine loads are not discussed. It is necessary that the authors also describe whether the presented results are average values from all samples from the tested engines. This would imply mixing of samples representative for different engine loads.

The abbreviations of ship types appear inconsistent and subjectively applied. Inconsistent in the sense that two types are named by installed power (high or low) and a third by fuel (HFO). Further, it is difficult to follow the specific ship name abbreviations. I suggest to use a different naming for ships that makes it easy to relate names to the relevant investigated category. An argumentation to support the cutoff in engine power between the chosen categories is necessary. It is not apparent to me why it should matter whether the engine is on a fishing vessel, a research vessel or an engineering vessel etc. Rather, categories should be based on whether it is a 4-stroke cycle or 2-stroke cycle and the engine speed. To use "diesel" as a term for distillate marine fuels can be confusing since diesel is mainly referred to when discussing engine types rather than fuels (also heavy fuel oil is used in marine diesel engines). "Marine distillate oil" could replace "diesel" to avoid the confusion on whether a fuel or engine type is discussed.

The work includes a lot of literature references and reasoning about the particle number size distributions, which could possibly be cut down to make the manuscript more concise. No own results of PN is presented.

Influence of measurement methods can be very important. I strongly recommend that the sampling system, the dilution ratios and the fuels used are described at least in the supplementary material. I have not had a chance to review this.

I find that in the results and discussion section it can be difficult to understand what is findings from this study, and what was previously known. E.g. page 16 lines 371-390 and page 21 line 501 to 505. I think this can be solved by changing the tense in the presentation.

Detailed: Abstract line 28 and 29 "basically presenting downward distribution trends with the increase of particle size" – difficult to understand, rewrite Abstract line 34 and 35. Conclusions on OC1, OC2 etc should be summarized in the abstract. The fractions of OC and EC need introduction given in the text but are not suitable for the abstract. Page 6 Line 140. Is 12 different vessels the same as 12 different engines? Clarify. Page 10 Line 238. As I read the referred to article the size distribution that results are compared with are from HFO combustion and not marine distillate oil "diesel". Page 10 Line 249 (and reoccurring) It is stated the the HFO ship has low combustion efficiency and elsewhere that 2-stroke engines have lower combustion efficiency than 4-stroke. Is this referring to the particular engine studied? Please, specify combustion efficiencies of the engines if this should be part of the explanations of the results. This also relates to the reasoning on page 11 Line 289 and below discussing sir/fuel ratios which should not differ much between an engine burning HFO and marine distillate oil. Please check your reference again. Page 10, line 259 and below. I suggest leave out since it seems this refers to a calculation rather than measurement. If not, marine aux engines are often of comparable power to the main engines in this study. Page 18, line 443. Misspelled "existing" Page 18, Line 445 "high volatile organic matter" should be either "highly volatile organic matter" or "high volatile organic matter content" or similar.. Page 25, Line 563. The ratio of HMW/LMW PAHs is said to indicate that fine particles are more toxic than coarse. Should it not be HMW/particle mass unit? I do not follow the reasons given.

---

## Author Comment (AC1) · 24 Oct 2019

Response to the comments

Journal: Atmospheric Chemistry and Physics

Manuscript ID: acp-2019-363

Title: Size-segregated characteristics of OC, EC and organic matters in PM emitted from different types of ships in China

Author(s): Fan Zhang, Hai Guo*, Yingjun Chen*, Volker Matthias, Yan Zhang, Xin Yang, Jianmin Chen Corresponding author: Hai Guo (hai.guo@polyu.edu.hk)

[Figure]

Yingjun Chen (yjchenfd@fudan.edu.cn)

The constructive comments of the reviewers are highly appreciated. We have revised the manuscript accordingly. Our point-by-point responses (in black) to each comment are listed below. And the modifications in the revised manuscript are marked in red. Please see the manuscript for details.

Reply to Referee 1#

**I lack a discussion on the relevance of the study in a larger context. It should be elaborated upon from the viewpoint that the engines are relatively small, high speed marine engines, and only representative for smaller marine craft.**

Response:

The comment is well noted. Explanation about the studied vessels has been added in Section 2.1 according to the comment. Most of the studied vessels in this study were indeed with high-speed marine engines. However, YK was a large ocean-going low-speed engine ship with heavy fuel oil used, while DFH was a medium-speed engine ship with marine diesel used (See Table S1 for details). Hence, we considered that they were representative of large/medium marine crafts in China to some extent.

Revision was made as follows (Lines 154-156, Section 2.1):

"Notably, most of the vessels had high-speed marine engines in this study, except for YK and DFH, which had low-speed engine and medium-speed engine, respectively."

**I find that very little attention is given to the engine and combustion characteristics. It is stated that "Overall, fuel type, fuel quality, engine type might have higher influence on particle mass distributions from ships than the operating mode", but it seems from Figure 1 that also operational mode is important for individual ships. Results on EC and OC at different engine loads are not discussed. It is necessary that the authors also describe whether the presented results are average values from all samples from the tested engines. This would imply mixing of samples representative for different engine**

[Figure]

loads. #

Response:

Thanks for pointing out these. More parameters of the tested engines have been added and shown in Table S1. The influence of combustion efficiency on the size distribution, organic formation, and PAHs distribution from different types of ships was briefly discussed in this manuscript in lines 248 to 256, lines 292to 297, lines 385 to 388, etc. Since all the tests were onboard (in-situ) measurements, rather than the bed tests that could control all the conditions in this study, we could not draw absolute conclusions about the engine and combustion characteristics here. Therefore, inferences caused by comprehensive influence factors were given and discussed in this study. Further study of a bed test for a low-speed HFO engine is in progress, from which we hope we can get more detailed and direct influence characteristics.

As for the size distributions of particles in different operation modes, we did not obtain an absolute conclusion before because not all the ships showed consistent size distribution patterns in different operation modes. However, more coarse particles were indeed found in operating modes with higher engine loads. According to the comments of the reviewer, the sentence has been revised in lines 257 to 264 in the revised manuscript as follows: "The size-segregated particle mass distributions in different operating modes were also compared in this study. The results showed that when the ships were operated with higher engine loads, there were more particles in coarse modes for most of the ships, which was similar to a previous study on mass distribution from measurements onboard of three ships (Fridell et al., 2008), but different from another study on a marine engine that reported the particle mass distribution centered at 0.1-0.2 $\mu$m with much fewer coarse particles under at-berth condition compared to maneuvering and ocean-going conditions (Chu-Van et al., 2017)." The conclusion has also been improved as "Overall, fuel type, fuel quality, engine type, and operating mode might have comprehensive influence on particle mass distributions from ships." (Lines 269 to 271 in the revised manuscript).

Actually, emission factors and the ratios of OC and EC at different engine loads have been discussed in our previous studies (Zhang et al., 2016; 2018). In this study, we focused on the size-segregated OC and EC distributions. Therefore, no further discussion of OC and EC at different engine loads was given here. Besides, due to the relatively small sampling size of size-segregated samples under different engine loads, not all the operating conditions could be included in the onboard tests. Hence, no solid conclusion could be drawn. However, as we mentioned above, a bed test for a low-speed HFO engine is in progress under different engine loads (0%, 25%, 50%, 75%, and 90%). We hope that we could get detailed and robust results of OC and EC distributions at different engine loads from this bed test.

At last, we would like to clarify that the current results are obtained based on the average values of all samples from the tested engines. Relative contents have been added in lines 275 to 276 and lines 472 to 474 in the revised manuscript.

"Notably, OC and EC levels given in this study were the average values of all samples that were classified into these three types."

"Same as OC and EC, the levels of PAHs and n-alkanes provided in this study were the average values of all samples that were classified into these three types."

**The abbreviations of ship types appear inconsistent and subjectively applied. Inconsistent in the sense that two types are named by installed power (high or low) and a third by fuel (HFO). Further, it is difficult to follow the specific ship name abbreviations. I suggest to use a different naming for ships that makes it easy to relate names to the relevant investigated category. An argumentation to support the cutoff in engine power between the chosen categories is necessary. It is not apparent to me why it should matter whether the engine is on a fishing vessel, a research vessel or an engineering vessel etc. Rather, categories should be based on whether it is a 4-stroke cycle or 2-stroke cycle and the engine speed. To use "diesel" as a term for distillate marine fuels can be confusing since diesel is mainly referred to when discussing engine types**

rather than fuels (also heavy fuel oil is used in marine diesel engines). "Marine distillate oil" could replace "diesel" to avoid the confusion on whether a fuel or engine type is discussed. #

Response:

Thank you for pointing out these confusions. Firstly, according to the reviewer's suggestion, we added the engine type (2-stroke or 4-stroke) in Table S1. YK was a 2-stroke low-speed engine ship, and the others were 4-stroke engine ships (1 medium-speed engine and 10 high-speed engines). Among the 4-stroke engine ships, fishing boats were classified as a special type, because they usually had much smaller engine power with older engine, and used non-standard diesel with much larger quantities (6 times the amount of water transport vessels in China) than other water transport vessels. Secondly, the fishing boats with engine power less than 300 kW could account for 75.6% of the total fishing boats in China, and the China registered ships with engine power less than 250 kW could account for 71.6% of the total registered ships in China. Therefore, combined with the studied ships, we considered 300 kW as a threshold between high engine power and low engine power in this study. Thirdly, fuel type is also a significant influence factor for the emitted pollutants. YK used typical heavy fuel oil as fuel, and the fishing boats used non-standard diesel as fuel, while the other ships used typical marine distillate oil as fuel. Therefore, after comprehensive consideration of the engine type, engine power and fuel type, we classified the studied ships into three categories in this study. We renamed the categories and abbreviations accordingly, namely, 4-stroke low-power diesel fishing boat (4-LDF), 4-stroke high-power marine-diesel vessel (4-HMV) and 2-stroke high-power heavy-fuel-oil vessel (2-HHV). All the abbreviations have been changed in the revised manuscript. Finally, due to the fact that our research on shipping emissions was a complete series, including the emission factors and characteristics of gaseous pollutants, PM and its chemical composition, the influence factor, the organic matters, and the total emissions in our previously published papers (Zhang et al., 2016, 2018, 2019), we would like to retain the specific ship

name abbreviations in this study for easy follow-up.

The "diesel" has been changed to "marine distillate oil" or "diesel fuel" in the revised manuscript according to the helpful comments.

**The work includes a lot of literature references and reasoning about the particle number size distributions, which could possibly be cut down to make the manuscript more concise. No own results of PN is presented.**

Response:

We thank the reviewer for his/her valuable comments. The contents and literature references in lines 217 to 222 in the original manuscript have been deleted. The reasoning about the particle number size distributions was also removed in lines 338 to 339 in the original manuscript. The description was also improved in lines 358 to 365 in the revised manuscript.

"In these processes, several factors can influence the size of the emitted particles: 1) surface growth involves the attachment of gas-phase species to the existing particles and results in increase of PM diameter and mass; 2) aggregation leads to chains and clusters of primary soot particles and an increase in particle size (to 10-100 nm in diameter) (Heywood, 1988); and 3) oxidation occurs during the formation and growth processes of the particles, resulting in the formation of gaseous species and reduction of soot particles and their precursors."

**Influence of measurement methods can be very important. I strongly recommend that the sampling system, the dilution ratios and the fuels used are described at least in the supplementary material. I have not had a chance to review this.**

Response:

Thanks for the comments. In this study, the size-segregated particle samples were collected directly from onboard tests through two sampling systems shown in the following Figures R1 and R2. It is noteworthy that these two sampling systems were carefully

introduced in our previous studies (Zhang et al., 2016, 2018). Therefore, we did not put them in the supplementary material in this study. Please note, sampling system (I) was used for the high-power diesel vessels of HH, DFH, XYH, and sampling system (II) was used for the other vessels. The difference of the two sampling systems was that the sampling system (I) had no dilution system due to the limitation of the sampling system used in our early sampling period, and consequently all the particles were collected directly from the flue gas. Dilution ratios for all the tested ships ranged from 1.0 to 4.2 according to the actual sampling conditions, which were added in Table S3 in the supporting material. Moreover, the detailed parameters of all the fuels used for each tested vessel are given in Table S2. Some descriptions about the fuels were also added in lines 158-160 in Section 2.1.

"Marine distillate oil was used as the fuel of the high-power-diesel vessels. Non-standard marine distillate oil was used for the 4-stroke low-power diesel fishing boats, which was introduced in our previous study (Zhang et al., 2016), while heavy fuel oil was used for the tested HFO vessel in this study."

Figure R1 Sampling system (I) of the onboard test

Figure R2 Sampling system (II) of the onboard test

**I find that in the results and discussion section it can be difficult to understand what is findings from this study, and what was previously known. E.g. page 16 lines 371-390 and page 21 line 501 to 505. I think this can be solved by changing the tense in the presentation.**

Response:

Sorry for the confusion. The tense was changed in some sentences, and some descriptions were improved in order to distinguish the findings of this study from previous studies. Revisions were shown in lines 371 to 390 and lines 499 to 504 in the revised manuscript.

"Soot particles are mainly formed through pyrolysis of diesel fuel and lubrication oil, and the organic fraction is formed through incomplete combustion of fuel and lubrication oil (Zetterdahl, 2016). Incomplete combustion of fuel and lubrication oil can significantly enhance the formation of fine particles, which might be the reason for the higher OC to EC ratio in particles with Dp < 0.43 $\mu$m in this study. This study also found that most EC contents were in the particles with Dp < 1.1 $\mu$m, coincident with the fact that soot is primarily in accumulation mode with particles of 0.1 < Dp <1 $\mu$m (Kasper et al., 2007). Although OC was also concentrated in fine particles with Dp<1.1 $\mu$m (Section 3.1), its percentage was higher in particles with Dp<0.43 $\mu$m, and lower in particles with 0.43<Dp<1.1 $\mu$m, compared to the EC distribution. Therefore, OC to EC ratios were the lowest in particles with 0.43<Dp<1.1 $\mu$m for all the tested ships. For the coarse particles in this study, though non-carbonous components were the dominant parts, OM increased after the cooling of the exhaust gas while content of EC and ash remained unchanged through adsorption of hydrocarbons and other volatile compounds (Zetterdahl, 2016), which enhanced the formation of coarse particles. Consequently, OC to EC ratios also showed high values in coarse particles with Dp>3.3$\mu$m. 2-HHV showed higher OC to EC ratio than the diesel fuel ships, which might be caused by the relatively lower combustion efficiency. Similarly, because 4-HMV ships had higher combustion efficiencies due to the higher fuel quality and better engine maintenance in this study, OC to EC ratios of 4-HMV showed the lowest levels."

And,

"The relatively longer exhaust gas channel and higher temperature in the channel of 2-HHV enhanced the formation of new secondary nucleation-mode and/or accumulation-mode particles from gas-phase organic compounds, leading to higher proportions of organic matters in fine particles, which has been proved in a previous study (Kittelson, 1998)."

References:

Zhang, F., Chen, Y. J., Tian, C. G., Lou, D. M., Li, J., Zhang, G., and Matthias, V.: Emission factors for gaseous and particulate pollutants from offshore diesel engine vessels in China, Atmos. Chem. Phys., 16, 6319-6334, 10.5194/acp-16-6319-2016, 2016.

Zhang, F., Chen, Y., Chen, Q., Feng, Y., Shang, Y., Yang, X., Gao, H., Tian, C., Li, J., Zhang, G., Matthias, V., and Xie, Z.: Real-World Emission Factors of Gaseous and Particulate Pollutants from Marine Fishing Boats and Their Total Emissions in China, Environ. Sci. Technol., 52, 4910-4919, 10.1021/acs.est.7b04002, 2018.

Zhang, F., Chen, Y., Cui, M., Feng, Y., Yang, X., Chen, J., Zhang, Y., Gao, H., Tian, C., Matthias, V., and Liu, H.: Emission factors and environmental implication of organic pollutants in PM emitted from various vessels in China, Atmos. Environ., 200, 302-311, https://doi.org/10.1016/j.atmosenv.2018.12.006, 2019.

Please also note the supplement to this comment:
https://www.atmos-chem-phys-discuss.net/acp-2019-363/acp-2019-363-AC1-supplement.zip

[Figure]

Figure R1 Sampling system (I) of the onboard test

**Fig. 1.** Figure R1

Figure R2 Sampling system (II) of the onboard test

**Fig. 2.** Figure R2

---

## Author Comment (AC2) · 24 Oct 2019

Response to the comments

Journal: Atmospheric Chemistry and Physics

Manuscript ID: acp-2019-363

Title: Size-segregated characteristics of OC, EC and organic matters in PM emitted from different types of ships in China

Author(s): Fan Zhang, Hai Guo*, Yingjun Chen*, Volker Matthias, Yan Zhang, Xin Yang, Jianmin Chen

[Figure]

Corresponding author: Hai Guo (hai.guo@polyu.edu.hk) Yingjun Chen (yjchenfd@fudan.edu.cn)

The constructive comments of the reviewers are highly appreciated. We have revised the manuscript accordingly. Our point-by-point responses (in black) to each comment are listed below. And the modifications in the revised manuscript are marked in red. Please see the manuscript for details.

Reply to Referee 2#

1. # It's authors' duty to explain what's new in this paper compared to several other papers published by the same group, especially the one on Atmospheric Environment 2019. Both titles indicate similar contents. #

Response:

Thanks for your comment. The paper published on Atmospheric Environment 2019 was focusing on the emission factors, profiles and characteristics of organic matters from the total particles emitted from ships. This information could provide some basic data for inventory estimation, source apportionment, and implication for source identification and health influence of ship emissions. However, this manuscript is focusing on characteristics of OC, EC and organic matters from size-segregated particles. As shown in the manuscript, the proportions and characteristics of these compositions varied significantly in different particle sizes, which were worthy being analyzed carefully. This information could provide further knowledge of the composition of particles in different sizes, implication for particle formation mechanism of ship exhaust, and also potential health impact, source apportionment of particles in different sizes. For example: the different profiles of chemical components in size-segregated particles implied that size-segregated chemical profiles should be considered when source apportionment was conducted. Furthermore, this study found more toxic organics such as PAHs in small particles emitted from fishing boats, suggesting the necessity of more stringent control on this type of boats in China.

2. # The results were not organized in a clear way which made them very hard to follow. 1) For different particle size bins, what are the chemical component profiles? No figure gives a comprehensive component profile. Only the OC/EC percentages compared among bins were provided. Authors failed to touch the whole picture of the "size-segregated characteristics", which should be the most important part of this study. For example, EC and OC were found very low for coarse PM. Then, which components are the major part for coarse PM? In addition, without a total mass analysis, it's impossible to judge the reliability of the sampling and analysis. 2) If the whole picture of size based chemical profile could be provided, then it's OK to discuss the distribution for each component in different size bins. However, this information is not very important compared with part 1). Currently, authors spent too much pages on discussing this, including Fig. 2, 6 and etc. 3) All figures were displayed in percentage or ratio. The mass of OC, EC, PAH or others should be provided directly. Is it still necessary to provide so many figures if the mass could be given? 4) Figure 1 is confusing. 'YK, GB1...' should be replaced with ship types, e.g. 'HDPV'. #

Response:

The valuable comments are highly appreciated.

1) Unfortunately, we could not get the detailed chemical profiles of the size-segregated particles in this manuscript. We only focused on OC, EC, organic matters of 16 priority PAHs, and n-alkanes in the particles. Other inorganic matters such as water-soluble ions and metal elements were not analyzed because the sample volume was too low of the particles in each size bin, especially in coarse particles. We inferred that inorganic matters of ash and hydrated sulfates could be the dominant component in coarse particles, which has been confirmed by a previous study (Moldanová et al., 2009). The detailed chemical profiles of particles in different size bins still need further investigation, which is also a target we are working on. Besides, we compared the total PM concentrations between the two sampling methods, namely the TSP sampling method and the Andersen sampling method. The result showed that they had a Pearson correlation coefficient of 0.917, and the correction was significant at the 0.05 level. As described in Sections 2.2, 2.3 and 2.4, all the sampling processes and chemical analysis were carried out according to standard methods, which showed reliable quality assurance and quality control. Therefore, we believed the sampling and analysis were reliable.

2) As explained above, we could not quantify the size-segregated inorganic matters in this study. Instead, carbonaceous matters of OC, EC, PAHs and n-alkanes in the PM were the focus in this study. Even though the detailed chemical profiles of size-segregated PM were not able to give, the results were still meaningful for the implications of climate change, source apportionment, health influence, and formation mechanism of organic matters, as presented and discussed in this manuscript. We fully agree that the whole picture of size based chemical profile is very important and needs to be figured out correctly in the future.

3) Firstly, the emission factors of the total PM and size-resolved particle mass distributions are given in Figure 1 in the manuscript. It could be seen that the PM emission factors varied significantly among different ships, from 0.08 to 19.01 g (kg fuel)-1. When the mass was distributed to different size bins, there were still large variations among the different types of ships (such as the OC and EC emission factors in different particle size bins shown in the following Figure R1). Due to the significant variations of the absolute mass/emission factor, it would be difficult to obtain the common pattern of the distributions and characteristics of OC, EC, and organic matters. Therefore, figures in percentage or ratio were displayed in this manuscript. Besides, since the distribution patterns of total OC, EC, PAHs and n-alkanes in different particle size bins are different, we consider that figures such as Figure 2 and Figure 6 are necessary in the manuscript.

Figure R1 OC and EC emission factors in different particle size bins

4) Figure 1 has been improved in the revised manuscript, and 2-HHV, 4-HMV, and

4-LDF have been added in the figure (shown as follows).

Figure 1 Emission factor for total PM and size-resolved particle mass distributions with different modes for the 12 tested ships

3. # Presentation quality in text also needs to be improved. 1) In abstract, line 30, 'in fine particles, OC and EC were the dominant components'. line 34, 'OC and EC have the lowest values for 0.43 to 1.1 um'. Are they still dominant? 2)Line 34, What are the OC1, OC2 and OC3? 3) Line 214, how can 5% be called the large proportion? 4) line 282, HFOV vessels should be HFOV. and 'HPDV ships' should be 'HPDV'. And this sentence is confusing. What's the meaning by 'HPDV accounted for 23%....'? Compared with what kind of ships? #

Response:

Thank you for pointing out these. OC and EC were indeed the dominant components in fine particles. In line 34, we meant that the OC to EC ratios had the lowest values for particles between 0.43 and 1.1 $\mu$m. They are not contradictory.

1) As explained in lines 392 to 397: OC and EC were tested according to the IMPROVE-A protocol in the thermal-optical carbon analysis. OC1, OC2, OC3, OC4, EC1, EC2, EC3, and pyrolysis carbon fragments were tested under different temperatures and conditions, which could be read directly from the result file (shown in the following Figure R2 as an example). Then OC and EC could be calculated according to the protocol. We analyzed the OC1 to EC3 fragments in this manuscript to help understand the different formation processes of particles.

Figure R2 Example of test results for OC and EC fragments

2) Sorry for the confusion. "large proportions" has been changed to "non-ignorable proportions" in the revised manuscript in line 221.

3) "HFOV vessels" has been revised to "2-HHV", and "HPDV ships" has been changed to "4-HMV" in the revised manuscript in lines 285 to 290. The abbreviations of ship

types have been changed to 4-stroke low-power diesel fishing boat (4-LDF), high-power-diesel vessel (4-HMV) and 2-stroke high-power heavy fuel oil vessel (2-HHV) (see Table S1 for details).This sentence has been improved as "For example, 4-HMV only accounted for 23% OC and 27% EC in particles with Dp < 0.43 $\mu$m compared to 2-HHV which had 75% OC and 66% EC in particles with Dp < 0.43 $\mu$m. This is in accordance with the characteristics of total PM mass distributions; that is, diesel fuel vessels had relatively smaller proportions of fine particles with Dp<0.43 $\mu$m and larger proportions of coarse mode particles than HFO ships."

Reference:

Moldanová, J., Fridell, E., Popovicheva, O., Demirdjian, B., Tishkova, V., Faccinetto, A., and Focsa, C.: Characterisation of particulate matter and gaseous emissions from a large ship diesel engine, Atmos. Environ., 43, 2632-2641, 10.1016/j.atmosenv.2009.02.008, 2009.

Please also note the supplement to this comment:
https://www.atmos-chem-phys-discuss.net/acp-2019-363/acp-2019-363-AC2-supplement.zip
* * *
[Figure]

Figure R1 OC and EC emission factors in different particle size bins

**Fig. 1.** Figure R1

[Figure]

Figure 1 Emission factor for total PM and size-resolved particle mass distributions
with different modes for the 12 tested ships

**Fig. 2.** Figure 1

```
* * *
         Peak Area          Carbon
OC1    OC      94  mv-secs  0.17  ug C/cm2    .17   ug C/filter
OC2    OC     764  mv-secs  1.37  ug C/cm2   1.37   ug C/filter
OC3    OC     621  mv-secs  1.11  ug C/cm2   1.11   ug C/filter
OC4    OC     539  mv-secs  0.97  ug C/cm2    .97   ug C/filter
EC1    EC     388  mv-secs  0.70  ug C/cm2    .70   ug C/filter
EC2    EC      16  mv-secs  0.03  ug C/cm2    .03   ug C/filter
EC3    EC       0  mv-secs  0.00  ug C/cm2    .00   ug C/filter
LRPyMin  Py     0  mv-secs   .00  ug C/cm2    .00   ug C/filter
LRPyMid  Py     0  mv-secs   .00  ug C/cm2    .00   ug C/filter
LRPyMax  Py     0  mv-secs   .00  ug C/cm2    .00   ug C/filter
LTPyMin  Py     0  mv-secs   .00  ug C/cm2    .00   ug C/filter
LTPyMid  Py     0  mv-secs   .00  ug C/cm2    .00   ug C/filter
LTPyMax  Py     0  mv-secs   .00  ug C/cm2    .00   ug C/filter
* * *
```

Figure R2 Example of test results for OC and EC fragments

**Fig. 3.** Figure R2

---

## Author Response (ED1)

**Response to the comments**

**Journal:** Atmospheric Chemistry and Physics

**Manuscript ID:** acp-2019-363

**Title:** Size-segregated characteristics of OC, EC and organic matters in PM emitted from different types of ships in China

**Author(s):** Fan Zhang, Hai Guo*, Yingjun Chen*, Volker Matthias, Yan Zhang, Xin Yang, Jianmin Chen

**Corresponding author:** Hai Guo (hai.guo@polyu.edu.hk)

Yingjun Chen (yjchenfd@fudan.edu.cn)

The constructive comments of the reviewers are highly appreciated. We have revised the manuscript accordingly. Our point-by-point responses (in black) to each comment are listed below. And the modifications in the revised manuscript are marked in red. Please see the manuscript for details.

Reply to Referee 1#

**I lack a discussion on the relevance of the study in a larger context. It should be elaborated upon from the viewpoint that the engines are relatively small, high speed marine engines, and only representative for smaller marine craft.**

**Response:**

The comment is well noted. Explanation about the studied vessels has been added in Section 2.1 according to the comment. Most of the studied vessels in this study were indeed with high-speed marine engines. However, YK was a large ocean-going low-speed engine ship with heavy fuel oil used, while DFH was a medium-speed engine ship with marine diesel used (See Table S1 for details). Hence, we considered that they were representative of large/medium marine crafts in China to some extent.

Revision was made as follows (Lines 154-156, Section 2.1):

"Notably, most of the vessels had high-speed marine engines in this study, except for YK and DFH, which had low-speed engine and medium-speed engine, respectively."

 # I find that very little attention is given to the engine and combustion characteristics. It is stated that "Overall, fuel type, fuel quality, engine type might have higher influence on particle mass distributions from ships than the operating mode", but it seems from Figure 1 that also operational mode is important for individual ships. Results on EC and OC at different engine loads are not discussed. It is necessary that  the authors also describe whether the presented results are average values from all samples from the tested engines. This would imply mixing of samples representative for different engine loads. #

**Response:**

Thanks for pointing out these. More parameters of the tested engines have been  added and shown in Table S1. The influence of combustion efficiency on the size distribution, organic formation, and PAHs distribution from different types of ships was briefly discussed in this manuscript in lines 248 to 256, lines 292to 297, lines 385 to 388, etc. Since all the tests were onboard (in-situ) measurements, rather than the bed tests that could control all the conditions in this study, we could not draw absolute  conclusions about the engine and combustion characteristics here. Therefore, inferences caused by comprehensive influence factors were given and discussed in this study. Further study of a bed test for a low-speed HFO engine is in progress, from which we hope we can get more detailed and direct influence characteristics.

As for the size distributions of particles in different operation modes, we did not  obtain an absolute conclusion before because not all the ships showed consistent size distribution patterns in different operation modes. However, more coarse particles were indeed found in operating modes with higher engine loads. According to the comments of the reviewer, the sentence has been revised in lines 257 to 264 in the revised manuscript as follows: "The size-segregated particle mass distributions in different  operating modes were also compared in this study. The results showed that when the ships were operated with higher engine loads, there were more particles in coarse modes for most of the ships, which was similar to a previous study on mass distribution from measurements onboard of three ships (Fridell et al., 2008), but different from another

study on a marine engine that reported the particle mass distribution centered at 0.1-0.2 µm with much fewer coarse particles under at-berth condition compared to maneuvering and ocean-going conditions (Chu-Van et al., 201⬚),

The conclusion has also been improved as "Overall, fuel type, fuel quality, engine type, and operating mode might have comprehensive influence on particle mass distributions from ships." (Lines 269 to 271 in the revised manuscript).

Actually, emission factors and the ratios of OC and EC at different engine loads have been discussed in our previous studies (Zhang et al., 2016; 2018). In this study, we focused on the size-segregated OC and EC distributions. Therefore, no further discussion of OC and EC at different engine loads was given here. Besides, due to the relatively small sampling size of size-segregated samples under different engine loads, not all the operating conditions could be included in the onboard tests. Hence, no solid conclusion could be drawn. However, as we mentioned above, a bed test for a low-speed HFO engine is in progress under different engine loads (0%, 25%, 50%, 75%, and 90%). We hope that we could get detailed and robust results of OC and EC distributions at different engine loads from this bed test.

At last, we would like to clarify that the current results are obtained based on the average values of all samples from the tested engines. Relative contents have been added in lines 275 to 276 and lines 472 to 474 in the revised manuscript.

"Notably, OC and EC levels given in this study were the average values of all samples that were classified into these three types."

"Same as OC and EC, the levels of PAHs and n-alkanes provided in this study were the average values of all samples that were classified into these three types."

**The abbreviations of ship types appear inconsistent and subjectively applied. Inconsistent in the sense that two types are named by installed power (high or low) and a third by fuel (HFO). Further, it is difficult to follow the specific ship name abbreviations. I suggest to use a different naming for ships that makes it easy to relate names to the relevant investigated category. An argumentation to support the cutoff in**

engine power between the chosen categories is necessary. It is not apparent to me why it should matter whether the engine is on a fishing vessel, a research vessel or an engineering vessel etc. Rather, categories should be based on whether it is a 4-stroke cycle or 2-stroke cycle and the engine speed. To use "diesel" as a term for distillate marine fuels can be confusing since diesel is mainly referred to when discussing engine types rather than fuels (also heavy fuel oil is used in marine diesel engines). "Marine distillate oil" could replace "diesel" to avoid the confusion on whether a fuel or engine type is discussed. #

**Response:**

Thank you for pointing out these confusions. Firstly, according to the reviewer's suggestion, we added the engine type (2-stroke or 4-stroke) in Table S1. YK was a 2-stroke low-speed engine ship, and the others were 4-stroke engine ships (1 medium-speed engine and 10 high-speed engines). Among the 4-stroke engine ships, fishing boats were classified as a special type, because they usually had much smaller engine power with older engine, and used non-standard diesel with much larger quantities (6 times the amount of water transport vessels in China) than other water transport vessels. Secondly, the fishing boats with engine power less than 300 kW could account for 75.6% of the total fishing boats in China, and the China registered ships with engine power less than 250 kW could account for 71.6% of the total registered ships in China. Therefore, combined with the studied ships, we considered 300 kW as a threshold between high engine power and low engine power in this study. Thirdly, fuel type is also a significant influence factor for the emitted pollutants. YK used typical heavy fuel oil as fuel, and the fishing boats used non-standard diesel as fuel, while the other ships used typical marine distillate oil as fuel. Therefore, after comprehensive consideration of the engine type, engine power and fuel type, we classified the studied ships into three categories in this study. We renamed the categories and abbreviations accordingly, namely, 4-stroke low-power diesel fishing boat (4-LDF), 4-stroke high-power marine-diesel vessel (4-HMV) and 2-stroke high-power heavy-fuel-oil vessel (2-HHV). All the abbreviations have been changed in the revised manuscript.

Finally, due to the fact that our research on shipping emissions was a complete series, including the emission factors and characteristics of gaseous pollutants, PM and its chemical composition, the influence factor, the organic matters, and the total emissions in our previously published papers (Zhang et al., 2016, 2018, 2019), we would like to retain the specific ship name abbreviations in this study for easy follow-up.

The "diesel" has been changed to "marine distillate oil" or "diesel fuel" in the revised manuscript according to the helpful comments.

**The work includes a lot of literature references and reasoning about the particle number size distributions, which could possibly be cut down to make the manuscript more concise. No own results of PN is presented.**

**Response:**

We thank the reviewer for his/her valuable comments. The contents and literature references in lines 217 to 222 in the original manuscript have been deleted. The reasoning about the particle number size distributions was also removed in lines 338 to 339 in the original manuscript. The description was also improved in lines 358 to 365 in the revised manuscript.

"In these processes, several factors can influence the size of the emitted particles: 1) surface growth involves the attachment of gas-phase species to the existing particles and results in increase of PM diameter and mass; 2) aggregation leads to chains and clusters of primary soot particles and an increase in particle size (to 10-100 nm in diameter) (Heywood, 1988); and 3) oxidation occurs during the formation and growth processes of the particles, resulting in the formation of gaseous species and reduction of soot particles and their precursors."

**Influence of measurement methods can be very important. I strongly recommend that the sampling system, the dilution ratios and the fuels used are described at least in the supplementary material. I have not had a chance to review this**

**Response:**

Thanks for the comments. In this study, the size-segregated particle samples were collected directly from onboard tests through two sampling systems shown in the following Figures R1 and R2. It is noteworthy that these two sampling systems were carefully introduced in our previous studies (Zhang et al., 2016, 2018). Therefore, we did not put them in the supplementary material in this study. Please note, sampling system (I) was used for the high-power diesel vessels of HH, DFH, XYH, and sampling system (II) was used for the other vessels. The difference of the two sampling systems was that the sampling system (I) had no dilution system due to the limitation of the sampling system used in our early sampling period, and consequently all the particles were collected directly from the flue gas. Dilution ratios for all the tested ships ranged from 1.0 to 4.2 according to the actual sampling conditions, which were added in Table S3 in the supporting material. Moreover, the detailed parameters of all the fuels used for each tested vessel are given in Table S2. Some descriptions about the fuels were also added in lines 158-160 in Section 2.1.

"Marine distillate oil was used as the fuel of the high-power-diesel vessels. Non-standard marine distillate oil was used for the 4-stroke low-power diesel fishing boats, which was introduced in our previous study (Zhang et al., 2016), while heavy fuel oil was used for the tested HFO vessel in this study."

[Figure]

165

Figure R1 Sampling system (I) of the onboard test

[Figure]

Figure R2 Sampling system (II) of the onboard test

170 # I find that in the results and discussion section it can be difficult to understand what is findings from this study, and what was previously known. E.g. page 16 lines 371-390 and page 21 line 501 to 505. I think this can be solved by changing the tense in the presentation. #

**Response:**

175 Sorry for the confusion. The tense was changed in some sentences, and some descriptions were improved in order to distinguish the findings of this study from previous studies. Revisions were shown in lines 371 to 390 and lines 499 to 504 in the revised manuscript.

"Soot particles are mainly formed through pyrolysis of diesel fuel and lubrication

180 oil, and the organic fraction is formed through incomplete combustion of fuel and lubrication oil (Zetterdahl, 2016). Incomplete combustion of fuel and lubrication oil can significantly enhance the formation of fine particles, which might be the reason for the

higher OC to EC ratio in particles with $D_p < 0.43$ μm in this study. This study also found that most EC contents were in the particles with $D_p < 1.1$ μm, coincident with the fact that soot is primarily in accumulation mode with particles of $0.1 < D_p < 1$ μm (Kasper et al., 2007). Although OC was also concentrated in fine particles with $D_p < 1.1$ μm (Section 3.1), its percentage was higher in particles with $D_p < 0.43$ μm, and lower in particles with $0.43 < D_p < 1.1$ μm, compared to the EC distribution. Therefore, OC to EC ratios were the lowest in particles with $0.43 < D_p < 1.1$ μm for all the tested ships. For the coarse particles in this study, though non-carbonous components were the dominant parts, OM increased after the cooling of the exhaust gas while content of EC and ash remained unchanged through adsorption of hydrocarbons and other volatile compounds (Zetterdahl, 20 6) which enhanced the formation of coarse particles. Consequently, OC to EC ratios also showed high values in coarse particles with $D_p > 3.3$ μm. 2-HHV showed higher OC to EC ratio than the diesel fuel ships, which might be caused by the relatively lower combustion efficiency Similarly, because 4-HMV ships had higher combustion efficiencies due to the higher fuel quality and better engine maintenance in this study, OC to EC ratios of 4-HMV showed the lowest levels."

And,

"The relatively longer exhaust gas channel and higher temperature in the channel of 2-HHV enhanced the formation of new secondary nucleation-mode and/or accumulation-mode particles from gas-phase organic compounds, leading to higher proportions of organic matters in fine particles, which has been proved in a previous study (Kittelson, 1998

Reply to Referee 2#

1.  # It's authors' duty to explain what's new in this paper compared to several other papers published by the same group, especially the one on Atmospheric Environment 2019. Both titles indicate similar contents. #

210 **Response:** Thanks for your comment. The paper published on Atmospheric Environment 2019 was focusing on the emission factors, profiles and characteristics of organic matters from the total particles emitted from ships. This information could provide some basic data for inventory estimation, source apportionment, and implication for source identification and health influence of ship emissions. However,

215 this manuscript is focusing on characteristics of OC, EC and organic matters from size-segregated particles. As shown in the manuscript, the proportions and characteristics of these compositions varied significantly in different particle sizes, which were worthy being analyzed carefully. This information could provide further knowledge of the composition of particles in different sizes, implication for particle formation

220 mechanism of ship exhaust, and also potential health impact, source apportionment of particles in different sizes. For example: the different profiles of chemical components in size-segregated particles implied that size-segregated chemical profiles should be considered when source apportionment was conducted. Furthermore, this study found more toxic organics such as PAHs in small particles emitted from fishing boats,

225 suggesting the necessity of more stringent control on this type of boats in China.

2.  # The results were not organized in a clear way which made them very hard to follow. 1) For different particle size bins, what are the chemical component profiles? No figure gives a comprehensive component profile. Only the OC/EC percentages

230   compared among bins were provided. Authors failed to touch the whole picture of the "size-segregated characteristics", which should be the most important part of this study. For example, EC and OC were found very low for coarse PM. Then, which components are the major part for coarse PM? In addition, without a total mass analysis, it's impossible to judge the reliability of the sampling and analysis.

 2) If the whole picture of size based chemical profile could be provided, then it's OK to discuss the distribution for each component in different size bins. However, this information is not very important compared with part 1). Currently, authors spent too much pages on discussing this, including Fig. 2, 6 and etc. 3) All figures were displayed in percentage or ratio. The mass of OC, EC, PAH or others should

 be provided directly. Is it still necessary to provide so many figures if the mass could be given? 4) Figure 1 is confusing. 'YK, GB1...' should be replaced with ship types, e.g. 'HDPV'. #

**Response:**

The valuable comments are highly appreciated.

1) Unfortunately, we could not get the detailed chemical profiles of the size-segregated particles in this manuscript. We only focused on OC, EC, organic matters of 16 priority PAHs, and n-alkanes in the particles. Other inorganic matters such as water-soluble ions and metal elements were not analyzed because the sample volume was too low of the particles in each size bin, especially in coarse particles. We inferred that inorganic matters of ash and hydrated sulfates could be the dominant component in coarse particles, which has been confirmed by a previous study (Moldanová et al., 2009). The detailed chemical profiles of particles in different size bins still need further investigation, which is also a target we are working on. Besides, we compared the total PM concentrations between the two sampling methods, namely the TSP sampling method and the Andersen sampling method. The result showed that they had a Pearson correlation coefficient of 0.917, and the correction was significant at the 0.05 level. As described in Sections 2.2, 2.3 and 2.4, all the sampling processes and chemical analysis were carried out according to standard methods, which showed reliable quality assurance and quality control. Therefore, we believed the sampling and analysis were reliable.

2) As explained above, we could not quantify the size-segregated inorganic matters in this study. Instead, carbonaceous matters of OC, EC, PAHs and n-alkanes in the PM were the focus in this study. Even though the detailed chemical profiles of sizesegregated PM were not able to give, the results were still meaningful for the implications of climate change, source apportionment, health influence, and formation mechanism of organic matters, as presented and discussed in this manuscript. We fully agree that the whole picture of size based chemical profile is very important and needs to be figured out correctly in the future.

3) Firstly, the emission factors of the total PM and size-resolved particle mass distributions are given in Figure 1 in the manuscript. It could be seen that the PM emission factors varied significantly among different ships, from 0.08 to 19.01 g (kg fuel)$^{-1}$. When the mass was distributed to different size bins, there were still large variations among the different types of ships (such as the OC and EC emission factors in different particle size bins shown in the following Figure R1). Due to the significant variations of the absolute mass/emission factor, it would be difficult to obtain the common pattern of the distributions and characteristics of OC, EC, and organic matters. Therefore, figures in percentage or ratio were displayed in this manuscript. Besides, since the distribution patterns of total OC, EC, PAHs and n-alkanes in different particle size bins are different, we consider that figures such as Figure 2 and Figure re necessary in the manuscript.

[Figure]

Figure R1 OC and EC emission factors in different particle size bins

4) Figure 1 has been improved in the revised manuscript, and 2-HHV, 4-HMV, and 4-LDF have been added in the figure (shown as follows).

[Figure]

Figure 💬 emission factor for total PM and size-resolved particle mass distributions with different modes for the 12 tested ships

3. # Presentation quality in text also needs to be improved. 1) In abstract, line 30, 'in fine particles, OC and EC were the dominant components'. line 34, 'OC and EC have the lowest values for 0.43 to 1.1 um'. Are they still dominant? 2)Line 34, What are the OC1, OC2 and OC3? 3) Line 214, how can 5% be called the large proportion? 4) line 282, HFOV vessels should be HFOV. and 'HPDV ships' should be 'HPDV'.

*And this sentence is confusing. What's the meaning by 'HPDV accounted for 23%....'? Compared with what kind of ships? #*

**Response:**

Thank you for pointing out these. OC and EC were indeed the dominant components in fine particles. In line 34, we meant that the OC to EC ratios had the lowest values for particles between 0.43 and 1.1 μm. They are not contradictory.

1) As explained in lines 392 to 397: OC and EC were tested according to the IMPROVE-A protocol in the thermal-optical carbon analysis. OC1, OC2, OC3, OC4, EC1, EC2, EC3, and pyrolysis carbon fragments were tested under different temperatures and conditions, which could be read directly from the result file (shown in the following Figure R2 as an example). Then OC and EC could be calculated according to the protocol. We analyzed the OC1 to EC3 fragments in this manuscript to help understand the different formation processes of particles.

```
* * *
          Peak Area              Carbon
OC1     OC      94 mv-secs   0.17 ug C/cm2    .17  ug C/filter
OC2     OC     764 mv-secs   1.37 ug C/cm2   1.37  ug C/filter
OC3     OC     621 mv-secs   1.11 ug C/cm2   1.11  ug C/filter
OC4     OC     539 mv-secs   0.97 ug C/cm2    .97  ug C/filter
EC1     EC     388 mv-secs   0.70 ug C/cm2    .70  ug C/filter
EC2     EC      16 mv-secs   0.03 ug C/cm2    .03  ug C/filter
EC3     EC       0 mv-secs   0.00 ug C/cm2    .00  ug C/filter
LRPyMin Py       0 mv-secs    .00 ug C/cm2    .00  ug C/filter
LRPyMid Py       0 mv-secs    .00 ug C/cm2    .00  ug C/filter
LRPyMax Py       0 mv-secs    .00 ug C/cm2    .00  ug C/filter
LTPyMin Py       0 mv-secs    .00 ug C/cm2    .00  ug C/filter
LTPyMid Py       0 mv-secs    .00 ug C/cm2    .00  ug C/filter
LTPyMax Py       0 mv-secs    .00 ug C/cm2    .00  ug C/filter
* * *
```

Figure R2 Example of test results for OC and EC fragments

2) Sorry for the confusion. "large proportions" has been changed to "non-ignorable proportions" in the revised manuscript in line 221.

3) "HFOV vessels" has been revised to "2-HHV", and "HPDV ships" has been changed to "4-HMV" in the revised manuscript in lines 285 to 290. The abbreviations of ship types have been changed to 4-
[revised manuscript text omitted]

---

## Author Response (AR2)

**Response to the comments**

**Journal:** Atmospheric Chemistry and Physics

**Manuscript ID:** acp-2019-363

**Title:** Size-segregated characteristics of OC, EC and organic matters in PM emitted from different types of ships in China

**Author(s):** Fan Zhang, Hai Guo*, Yingjun Chen*, Volker Matthias, Yan Zhang, Xin Yang, Jianmin Chen

**Corresponding author:** Hai Guo (hai.guo@polyu.edu.hk)

Yingjun Chen (yjchenfd@fudan.edu.cn)

The constructive comments of the editor are highly appreciated. We have revised the manuscript accordingly. Our point-by-point responses (in black) to each comment are listed below. And the modifications in the revised manuscript are marked in red. Please see the manuscript for details.

**Throughout the paper, please use 'particle mass (or PM mass) size-distribution' instead of 'size-segregated particle mass distributions'**

**Response:**

Thanks for the comment. "size-segregated particle mass distributions" have been changed to "particle mass size-distribution" in the revised manuscript accordingly.

**The fact that all 4-HMV tests were done at different sampling conditions (without dilution) could have significantly influenced especially the measured OC and hence the EC/OC ratios and PM mass, but also EC is influenced by the dilution. The EF for PM mass are also the lowest measured. The potential influence of the different sampling methods needs to be highlighted when the different ship types are compared, especially in terms of OC/EC or PM mass.**

**Response:**

We thank the editor for the valuable comments. When PM mass, OC, EC and OC/EC are compared among different ship types, the potential influence of the different

sampling methods have been highlighted accordingly in the revised manuscript. Such as in line 166, lines 217-219, lines 229-231, lines 250-252, lines 262-266, and lines 409-413 that are showing as follows:

"Notably, the sampling system applied for the HFO vessel of YK was the same diluted system used for the fishing boats (see Fig. 1 for detail)."

"Notably, the 4-stroke high-power marine-diesel vessels used an undiluted sampling system, which was inferred as one reason for the lower PM emission factors (Robinson, 2010)."

"Figure 1 Emission factor for total PM and size-resolved particle mass distributions with different modes for the 12 tested ships (the vessels on the right light gray side used the undiluted system, and the rest vessels used the diluted system)"

"The different particle mass size-distributions for different types of ships were likely caused by the quality of fuel and its combustion efficiency in the engines, and also the sampling method."

"The use of undiluted sampling system could be one reason of the lower proportions of fine particles of 4-HMV, which has been proved that under the condition of diluted sampling system, the soluble organic fractions could pass from gas phase to particle phase to form new particles (Heywood, 1988; Kittelson, 1998)."

"Similarly, because 4-HMV ships had higher combustion efficiencies due to the higher fuel quality and better engine maintenance in this study, and also the influence of the use of undiluted sampling system that adverse to the transmission of gas-phase organic matters to particle phase (Heywood, 1988; Kittelson, 1998), OC to EC ratios of 4-HMV showed the lowest levels."

**In Fig. 2 and 6 in the text to the y-axes replace "... distribution IN different particle size bins (%)" into "... distribution BETWEEN different particle size bins (%)". There are several places in the text where the wording should be changed accordingly.**

**Response:**

Thanks for pointing out these. The related "in" in the figures and texts have been replaced to "between" in the revised manuscript accordingly.

**Some additional comments are inserted in your response document**
**1. Line 54: consequently throughout the paper: this is called particle mass (or PM mass) size-distribution**

**Response:**

Thanks for the comment. "size-segregated particle mass distributions" have been changed to "particle mass size-distribution" in the revised manuscript accordingly.

**2. Line 55: ship**

**Response:**

Thanks for the comment. The sentence has been improved as: "The particle mass size-distribution in different operating modes of different ships were also compared in this study." in lines 267-268 in the revised manuscript.

**3. Line 61: The part of the sentence referring to Chu-Van does not really reflect difference between engine loads (at berth typically auxiliary engines are used)**

**Response:**

The comment is well noted. The sentences have been improved as: "The particle mass size-distribution in different operating modes of different ships were also compared in this study. The results showed that when the ships were operated with higher engine loads, there were more particles in coarse modes for most of the ships, which was similar to a previous study on mass distribution from measurements onboard of three ships (Fridell et al., 2008). Besides, a previous study on a marine engine reported that the particle mass distribution centered at 0.1-0.2 μm with much fewer coarse particles under at-berth condition with auxiliary engine compared to maneuvering and ocean-going conditions (Chu-Van et al., 2017)." in lines 267-277 in the revised manuscript.

**4. Line 136: meaning condensation of semi-volatile species??**

**Response:**

Thanks for your comment. According to the explanation of Heywood (1988), the surface growth includes both adsorption and condensation. Adsorption involves the adherence of molecules of unburned hydrocarbons to the surface of the soot particles by chemical or physical forces. Condensation will occur whenever the vapor pressure of the gaseous hydrocarbon exceeds its saturated vapor pressure. And the hydrocarbons most likely to condense are those of low volatility.

**5. Line 193: this is still rather confusing, have you observer OM with and without cooling of the exhaust to see increase in OM?**

**Response:**

Thanks for your comment. Unfortunately, we didn't compare the OM difference directly between with and without cooling of the exhaust in this study. However, we find some evidence that can support this inference from a previous study. The literature from Heywood about particle formation has been added in lines 398-402 in the revised manuscript. A figure (Typical effect of dilution ratio on particulate mass emission and its partitioning between extractable and non-extractable fractions) from this book is showing as follows. It is known that in the standard particulate mass emission measurement process, the cooling of the exhaust occurs in a dilution tunnel which simulates approximately the actual atmospheric dilution process. Therefore, as explained in this book, the non-extractable fraction is the carbonaceous soot generated during combustion and is not affected by the dilution process. The bulk of the extractable fraction is acquired after the exhaust gas is mixed with dilution air. And both adsorption and condensation occur. Adsorption involves the adherence of molecules of unburned hydrocarbons to the surface of the soot particles by chemical or physical forces. This depends on the fraction of the available particle surface area occupied by hydrocarbons and on the partial pressure of the gaseous hydrocarbons that drives the adsorption process. As the dilution ratio increase from unity, the effect of decreasing temperature on the number of active sites dominates and the extractable fraction increases. At high dilution ratios, the sample temperature becomes insensitive to the dilution ratio but the decreasing hydrocarbon partial pressure causes the extractable

mass to fall again. Condensation will occur whenever the vapor pressure of the gaseous hydrocarbon exceeds its saturated vapor pressure. Increasing dilution decreases hydrocarbon concentrations and hence vapor pressure. However, the associated reduction in temperature does reduce the saturation pressure. High exhaust concentrations of hydrocarbons are the conditions where condensation is likely to be most significant, and the hydrocarbons most likely to condense are those of low volatility. Sources of low-volatility hydrocarbons are the high-boiling-point end of the fuel, unburned hydrocarbons that have been pyrolyzed but not consumed in the combustion process, and the lubricating oil. In general, the cooling of exhaust gas does promote the increase of the organic matters.

[Figure]

**FIGURE 11-52**
Typical effect of dilution ratio on particulate mass emission and its partitioning between extractable and nonextractable fractions.[79]

"For the coarse particles in this study, though non-carbonous components were the dominant parts, OM increased after the cooling of the exhaust gas while content of EC and ash remained unchanged through adsorption of hydrocarbons and other volatile compounds (Zetterdahl, 2016; Heywood, 1988), which enhanced the formation of coarse particles."

**6. Line 195: 4-LDF??**

**Response:**

Thanks for the comment. This sentence has been improved in the revised manuscript in lines 403-406.

"Consequently, OC to EC ratios also showed high values in coarse particles with

Dp>3.3μm. 2-HHV showed higher OC to EC ratio than the diesel fuel ships of both 4-HMV and 4-LDF, which might be caused by the relatively lower combustion efficiency and higher EC emissions of 4-LDF (Zhang, 2018)."

**7. Line 196: or higher EC emissions in case of the 4-LDF vessels?**

**Response:**

Thanks for your valuable comment. This sentence has been improved in the revised manuscript in lines 403-406. 4-LDF vessels indeed have higher EC emissions that was observed in our previous study, which also has been added in this sentence. "Consequently, OC to EC ratios also showed high values in coarse particles with Dp>3.3μm. 2-HHV showed higher OC to EC ratio than the diesel fuel ships of both 4-HMV and 4-LDF, which might be caused by the relatively lower combustion efficiency and higher EC emissions of 4-LDF (Zhang, 2018)."

**8. Line 204: higher temperatures in the exhaust are not favoring condensation**

**Response:**

The comment is well noted. The "higher temperatures" in the sentence has been removed, and the revised sentence in lines 524-528 in the manuscript is showing as follows:

"The relatively longer exhaust gas channel in the channel of 2-HHV enhanced the formation of new secondary nucleation-mode and/or accumulation-mode particles from gas-phase organic compounds, leading to higher proportions of organic matters in fine particles, which has been proved in a previous study (Kittelson, 1998)."

**9. Line 285: The ships sampled without dilution should be indicated in the figure as this affects especially the PM mass. Please, insert vertical gridlines so one could easier relate the PM mass emission factors to ships**

**Response:**

Thanks for the comment. Figure 1 has been improved to separate the different

sampling systems. And the corresponding descriptions have been added in line 166 in section 2.1 and also in the text below Figure 1.

[Figure]

Figure 1 Emission factor for total PM and size-resolved particle mass distributions with different modes for the 12 tested ships (the vessels on the right light gray side used the undiluted system, and the rest vessels used the diluted system)

**10. Line 284: between (sum over the size bins gives 100%)? # and**

**11. Line 314: between**

**Response:**

The comments are well noted. The related "in" in the figures and texts have been replaced to "between" in the revised manuscript accordingly.

**Response:**

Thanks for your comment. Some explanation has been added to the error bars in lines 354-355 in the revised manuscript as follows:

[revised manuscript text omitted]

---

## Author Response (AR3)

**Response to the comments**

**Journal:** Atmospheric Chemistry and Physics

**Manuscript ID:** acp-2019-363

**Title:** Size-segregated characteristics of OC, EC and organic matters in PM emitted from different types of ships in China

**Author(s):** Fan Zhang, Hai Guo*, Yingjun Chen*, Volker Matthias, Yan Zhang, Xin Yang, Jianmin Chen

**Corresponding author:** Hai Guo (hai.guo@polyu.edu.hk)

Yingjun Chen (yjchenfd@fudan.edu.cn)

The constructive comments of the editor are highly appreciated. We have revised the manuscript accordingly. Our point-by-point responses (in black) to each comment are listed below. And the modifications in the revised manuscript are marked in red. Please see the manuscript for details.

Line 25 # are presented #

**Response**

Thanks for the comment. The sentence has been improved as: "In this study, size-segregated distributions and characteristics of particle mass, organic carbon (OC), elemental carbon (EC), 16 EPA PAHs and 25 n-alkanes measured on board of 12 different vessels in China are presented." in lines 23-26 in the revised manuscript.

Line 29 # sorry, I don't understand formulation 'basically presenting downward distribution trends with the increase of particle size', do you mean that relative contribution of OC, EC PAH and alkanes to the size-segregated particle mass is decreasing with increase in particle size? I would recommend to re-formulate this. #

**Response**

The comment is well noted. The sentences have been improved as: "(1) More than half of the total particle mass, OC, EC, PAHs and n-alkanes were concentrated in fine particles with aerodynamic diameter ($D_p$)<1.1 μm for most of the tested ships. The relative contributions of OC, EC, PAH and alkanes to the size-segregated particle mass are decreasing with the increase of particle size." in lines 26-31 in the revised manuscript.

Line 34 # you need to briefly introduce here what is OC1, OC2, OC3 and EC1-EC3 (like out of 4 OC fragments and 3 EC fragments obtained in thermal-optical analysis ...) #

**Response**

Thanks for your comment. The sentence has been improved as: "(3) Out of the four OC fragments and three EC fragments obtained in thermal-optical analysis, OC1, OC2 and OC3 were the dominant OC fragments for all the tested ships, while EC1 and EC2 were the main EC fragment for ships running on heavy fuel oil (HFO) and marine diesel fuel, respectively; Different OC and EC fragments presented different distributions in different particle sizes." in lines 36-40 in the revised manuscript according to your comment.

Line 347 # what represents the error bar - standard deviation, uncertainty of analysis, percentile? #

**Response**

Thanks for the comment. Some explanation has been added to the error bars in lines 351-353 in the revised manuscript as follows: "the upper error bar represents the standard deviation of EC, and the lower error bar represents the standard deviation of OC".

Line 390 # non-carbonaceous compounds #

**Response**

The comment is well noted. All the "non-carbonous components" have been changed to "non-carbonaceous compounds" in the revised manuscript.

Line 393 # mass #

**Response**

The comment is well noted. The word "formation" has been changed to "mass" in line 398 in the revised manuscript according to your comment.

[revised manuscript text omitted]